# An NS-segment exonic splicing enhancer regulates influenza A virus replication in mammalian cells

Xiaofeng Huang[1,2], Min Zheng[1,2], Pui Wang[1,2], Bobo Wing-Yee Mok[1,2], Siwen Liu[1,2], Siu-Ying Lau[1], Pin Chen[1,2], Yen-Chin Liu[1,2], Honglian Liu[1,2], Yixin Chen[3], Wenjun Song[1,4], Kwok-Yung Yuen[1,2] & Honglin Chen[1,2,3]

Influenza virus utilizes host splicing machinery to process viral mRNAs expressed from both M and NS segments. Through genetic analysis and functional characterization, we here show that the NS segment of H7N9 virus contains a unique G540A substitution, located within a previously undefined exonic splicing enhancer (ESE) motif present in the NEP mRNA of influenza A viruses. G540A supports virus replication in mammalian cells while retaining replication ability in avian cells. Host splicing regulator, SF2, interacts with this ESE to regulate splicing of NEP/NS1 mRNA and G540A substitution affects SF2–ESE interaction. The NS1 protein directly interacts with SF2 in the nucleus and modulates splicing of NS mRNAs during virus replication. We demonstrate that splicing of NEP/NS1 mRNA is regulated through a *cis* NEP-ESE motif and suggest a unique NEP-ESE may contribute to provide H7N9 virus with the ability to both circulate efficiently in avian hosts and replicate in mammalian cells.

[1] State Key Laboratory for Emerging Infectious Diseases, Department of Microbiology, The University of Hong Kong, Hong Kong SAR, China. [2] Collaborative Innovation Center for Diagnosis and Treatment of Infectious Diseases, The University of Hong Kong, Hong Kong SAR, China. [3] National Institute of Diagnostics and Vaccine Development in Infectious Diseases, State Key Laboratory of Molecular Vaccinology and Molecular Diagnostics, School of Public Health, Xiamen University, Xiamen 361005, China. [4] Department of Biotechnology, College of Life Science and Technology, Jinan University, 601 Huangpu Avenue West, Guangzhou 510632, China. Correspondence and requests for materials should be addressed to H.C. (email: hlchen@hku.hk).

nfluenza virus utilizes viral polymerase for transcription and replication of the viral genome in the nucleus of infected cells, but remains dependent on host splicing machinery for splicing of viral messenger RNAs[1–4]. A recent study using various genome-wide screening approaches confirms the critical role of host splicing machinery in influenza virus replication[5]. Among the eight negative sense RNA segments of the influenza virus genome, both M and NS express unspliced and spliced transcripts, which differ in their viral functions. Expression of spliced M and NS mRNAs is highly regulated[4]. It is of interest how regulation of transcript splicing may affect virus replication efficiency and host adaptation.

The NS segment of influenza virus expresses two viral proteins, non-structural protein 1 (NS1) and nuclear export protein (NEP, also known as NS2), from unspliced and spliced NS mRNAs, respectively[6–8]. NS1 is one of the earliest proteins expressed in virus infection and has important functions in inhibition of host antiviral gene expression and in virus replication[9–14]. NEP, which is essential for virus replication, is expressed from the spliced form of NS mRNA at a later point during virus infection[6,8,15]. However, it is largely unknown how NS1/NEP expression is regulated during virus replication. Splicing of mRNA is considered to be one of the most complex processes in cells and is performed by the spliceosome complex and splicing regulators[16]. Through interaction with various cis elements such as intronic splicing enhancer, exonic splicing enhancer (ESE), exonic splicing silencer or intronic splicing silencer, splicing regulators direct the splicing of pre-mRNAs. Although the splicing sites in influenza virus M and NS mRNAs are well known, the cis elements that are recognized by splicing regulators in NS mRNAs are less clearly defined. The splicing regulator SF2/ASF is reported to recognize an ESE in the 3′-splicing site region of M2 mRNA to regulate the M1/M2 splicing process[1]. During influenza virus infection, NS1 protein functions to alter mRNA processing, to facilitate virus evasion of host antiviral immunity[13]. Reports suggest that NS1 protein may be involved in regulation of the spliced form of NS mRNA, NEP, but other studies argue against a role for NS1 in this process[10,17,18]. The NS1 and NEP proteins play different roles during virus infection and regulation of splicing of mRNA to favour either NS1 or NEP could significantly affect virus replication efficiency. Indeed, it is suggested that inefficient splicing of NS mRNAs of the pandemic A/Brevig Mission/1918/1 (H1N1) virus resulted in higher levels of the NS1 protein, contributing to its pathogenic properties[19]. NEP is required for coordination of viral ribonucleoprotein (RNP) nuclear export in the later stage of infection and it is suggested that virus adapts to utilize a poor 5′-splice site to allow slow expression of NEP early in virus infection[20]. These studies indicate that regulation of the NS splicing process plays a key role in virus replication.

In the past two decades, sporadic human infections with multiple different subtypes of avian influenza viruses have occurred. The rapid establishment of avian H7N9 virus endemicity in chickens, accompanied by continuous emergence of H7N9 human cases since 2013 (ref. 21), suggests that this virus possesses an unusual fitness for both circulation and human infection. There is strong interest in understanding the mechanism underpinning the ability of this virus to cause human infections and identification of residues that support replication in mammalian cells is important for surveillance of circulating strains. Previous studies revealed that the 2013 H7N9 virus is a reassortant generated with haemagglutinin and neuraminidase genes from viruses present in wild birds, with internal genes derived from avian H9N2 viruses circulating in domestic poultry[22]. Host adaptation of H9N2 virus has been extensively studied; H9N2 viruses are believed to serve as a critical backbone for the emergence of novel subtype viruses in poultry that can cause infection in humans[22–24]. Although most studies have focused on how molecular markers associated with receptor binding specificity and polymerase PB2 protein activity enable H7N9 virus infection in humans[25–28], little is known about the general fitness of other internal genes of 2013 H7N9 and their contribution to virus replication.

The here performed analysis of the 2013 H7N9 virus genome revealed a distinct E172K substitution in the NS1 protein, corresponding to nucleotide G540A. Further analysis showed that this substitution is inherited from H9N2 virus. Notably, the NS G540A genotype emerged in early 2000, becoming the dominant genotype among H9N2 viruses from 2012 onwards. We identified the G540A substitution as being located in an uncharacterized NEP ESE motif that is recognized and bound by the splicing regulator SF2. Acquired NS G540A substitution altered the ratio of NEP/NS1 mRNA in virus infections and enhanced H7N9 virus replication in mammalian cells and mice. It seems that splicing of NS mRNAs can be regulated by the NEP ESE and that mutation within this ESE may alter virus replication efficiency.

## Results

**The 2013 H7N9 virus contains a unique NS-G540A substitution.** To investigate adaptations in the H7N9 virus genome, we compared available sequences and found a unique substitution, G540A, in the NS segment of H7N9 virus (Fig. 1a). G540A caused an E172K amino acid change in the NS1 protein, but not in NEP. As H7N9 virus acquired internal genes from avian H9N2 virus, we surveyed H9N2 NS sequences in the public database and discovered that G540A, as well as an adjacent nucleotide mutation, U539C, emerged in early 2000; since 2009–2011, U539C/G540A became increasingly prevalent, being the predominant variant from 2012 onwards (Supplementary Fig. 1). These substitutions are rare among other influenza virus subtypes. It is natural to assume that the internal genes derived from this dominant strain of H9N2 virus strain have provided the optimal fitness required for continuous circulation of H7N9 in the field. One recent report that analysed H9N2 evolution in China found a distinct genotype, G57, which donated internal genes for the formation of the H7N9 strain[29]. Consistent with this report, phylogenetic analysis showed that all H9N2 G57 genotype viruses have the NS G540A substitution and form a distinct group (Supplementary Fig. 2a). To characterize the effect of NS G540A (NS1 E172K) on virus replication, we first tested one pair of H7N9 viruses constructed from the backbone of the 2013 H7N9 virus strain[28]. The mutant virus contained an NS A540G (K172E) mutation, reversed from the prevailing NS genotype at this position, with a view to disrupting any function specifically associated with G540A. Growth kinetics analysis showed that A540G mutation caused attenuation of H7N9 virus replication in A549 human lung cancer cells, but not in DF-1 chicken fibroblasts (Fig. 1b). As the H7N9 virus acquired internal genes from H9N2, we similarly constructed two additional pairs of wild type (WT) and A540G mutant viruses; one set with a complete H9N2 virus genome (H9N2), the other with internal genes from H7N9 virus and surface genes derived from H9N2 (rH9N2)[30]. Consistent with that observed with the H7N9 backbone, introduction of the A540G substitution caused attenuation of both rH9N2 and H9N2 virus replication in A549 cells, but not in DF-1 cells (Fig. 1b). To further demonstrate the general effect of gaining G540A substitution on virus replication in mammalian cells, we tested growth kinetics using recombinant virus containing either the NS segment from a H7N9 (540A), H9N2 (540G) or mutated H9N2-NS (540A) virus with the rest of

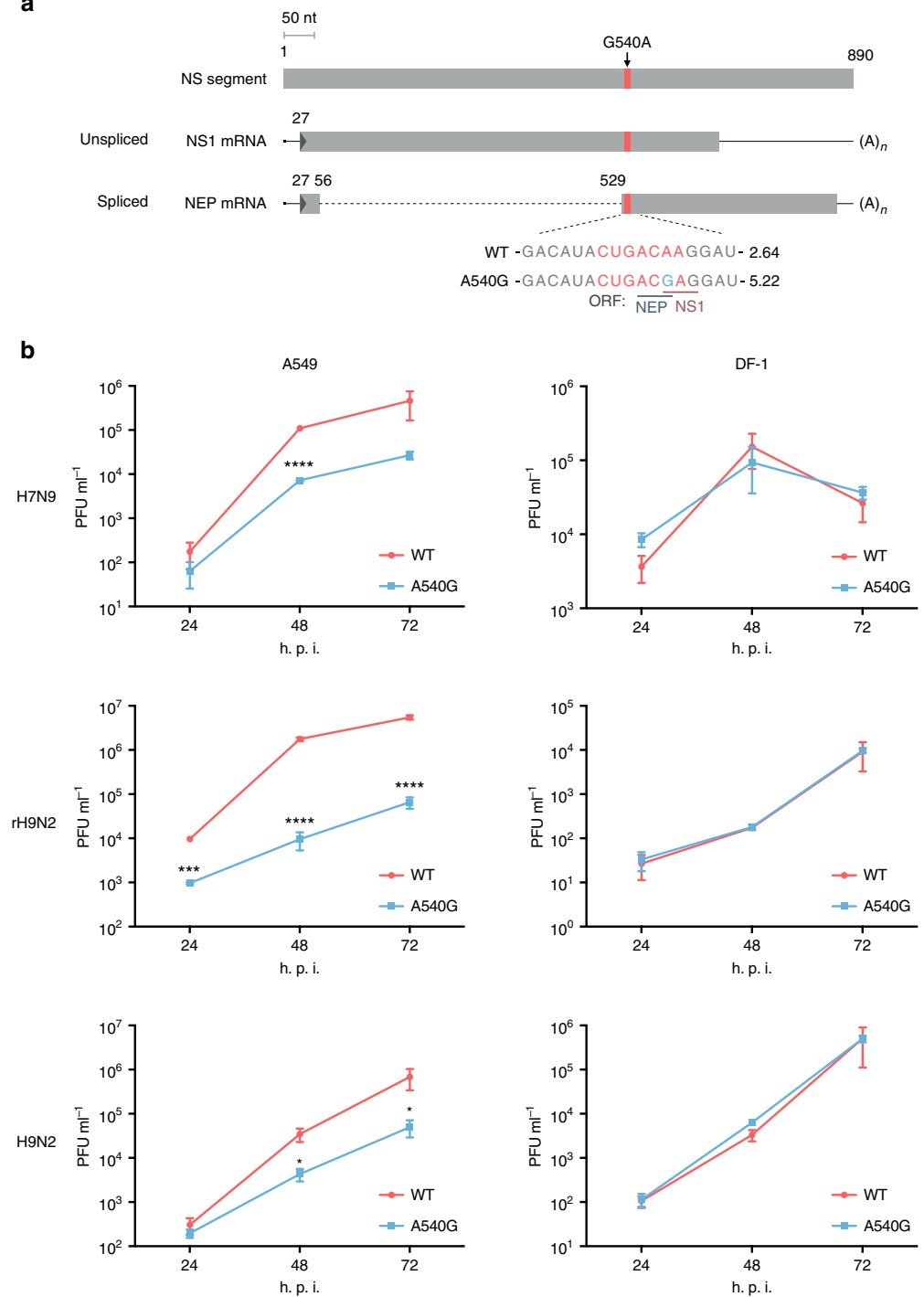

**Figure 1 | A540G nucleotide substitution in NS segment of H7N9 virus causes attenuated replication in human lung carcinoma cell line.** (**a**) Schematic illustration of NS transcripts and the putative ESE motif. Coding sequences are shown as a grey bar and the intron of NEP mRNA as a dotted line. The putative ESE site is shown in red and A540G is marked in blue. G540A substitution caused an amino acid change, E172K, in the NS1 protein and a silent mutation in the NEP. The SF2-ESE motif score calculated by the SF2/ASF matrix in the ESEfinder programme is shown on the right of the RNA sequence. (**b**) H7N9-WT[28], H7N9-NS-A540G, reassortant H9N2 (rH9N2), rH9N2-NS-A540G, WT H9N2 (A/HongKong/308/2014) or H9N2-NS-A540G viruses were rescued by the reverse genetics (RG) technique. The three pairs of H7N9, rH9N2 and H9N2 viruses only differ by the nucleotide substitution in NS, as indicated in **a**. The reassortant H9N2 and H9N2-NS-A540G (rH9N2) viruses contain haemagglutinin (HA) and neuraminidase (NA) surface genes from the A/HongKong/308/2014 (H9N2) strain, with the six internal genes being the same as those of the H7N9-WT and H7N9-NS-A540G viruses, respectively. A549 cells or DF-1 cells were infected with these RG viruses at a multiplicity of infection (MOI) of 0.01. Supernatants were harvested at the indicated time points and viruses titrated by plaque assay. Error bars represent mean ± s.d. ($n = 3$). Statistical significance was analysed by Student's $t$-test: *$P < 0.05$, ***$P < 0.001$ and ****$P < 0.0001$; h.p.i., hours post infection.

the segments from the A/WSN/33 (H1N1) backbone. The results showed that naturally acquired (H7N9) or artificially introduced (H9N2-E172K) G540A substitution in the NS segment significantly enhances virus replication in A549 cells (Supplementary Fig. 2b). As sequence surveillance found another substitution, C539U, which emerged alongside G540A in H9N2 viruses since 2000 and is present in all H7N9 viruses (Supplementary Fig. 1a), we also tested a mutant H7N9 virus containing a C539U back

mutation in the NS segment, both alone and in combination with A540G, and found that back mutation at position 539 did not affect virus replication in A549 cells (Supplementary Fig. 2c). The following experiments are therefore focused on characterizing the biological significance of the G540A mutation. The data suggest that G540A substitution in the NS segment provides H7N9 and H9N2 viruses with enhanced ability to replicate in mammalian cells, without reducing fitness in avian cells.

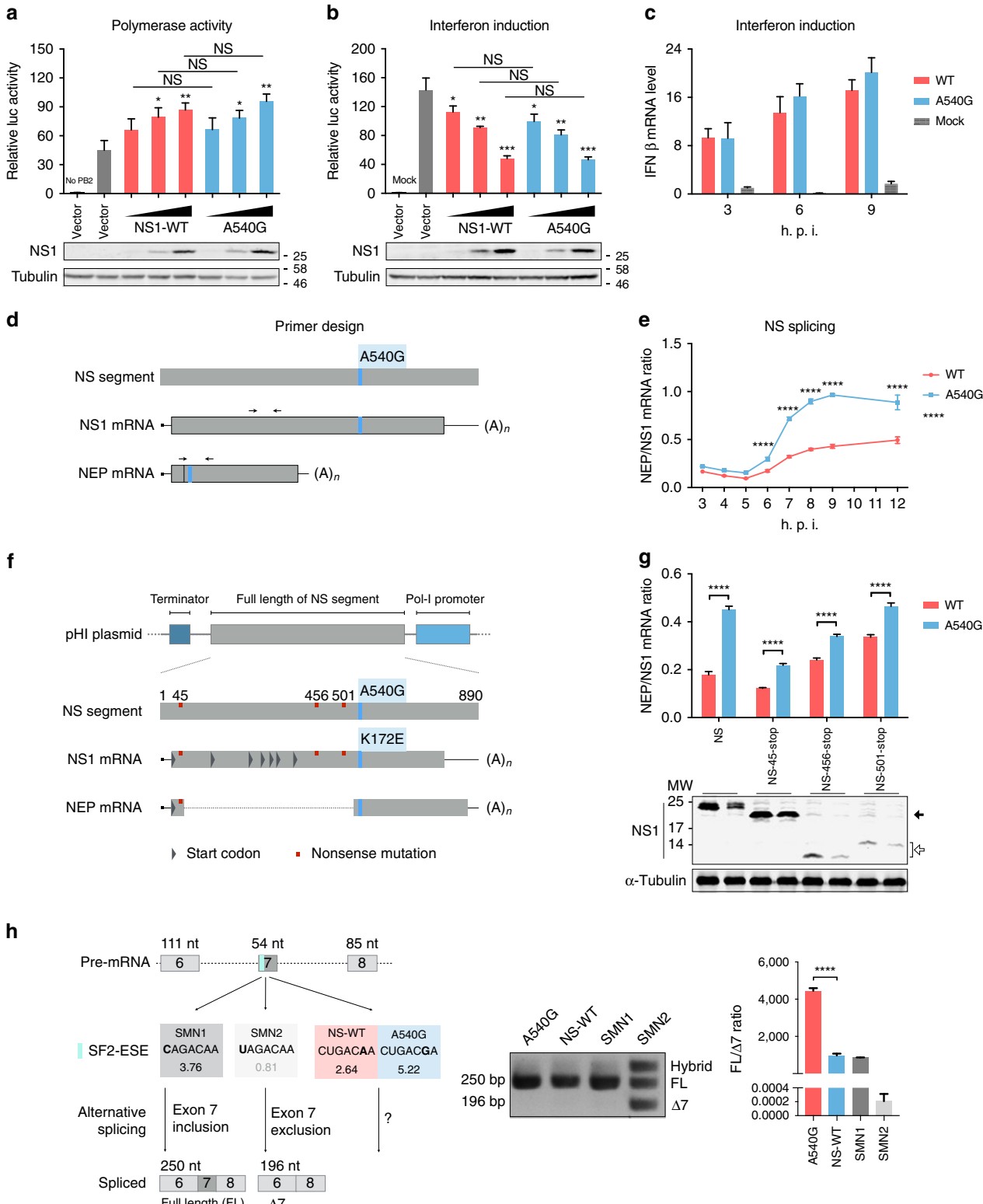

**Identification of an ESE site in exon 2 of NEP mRNA**. As the G540A substitution caused an amino acid change, E172K, in the NS1 protein but was a silent mutation in the NEP (Fig. 1a), we examined whether this alteration may alter NS1 function. The NS1 protein is known to interact with the viral polymerase complex and regulate viral RNP polymerase activity through interaction with host negative regulators[31,32]. We found that co-expression of increasing amounts of NS1 containing either 172E (540G) or 172K (540A) has no significantly different effect on RNP polymerase activity (Fig. 2a). The NS1 protein also inhibits expression of interferon (IFN)-β. However, we found that 172E-NS1 and 172K-NS1 both inhibit IFN-β reporter activity induced by Sendai virus infection similarly (Fig. 2b). We further tested IFN-β expression in virus-infected cells and found no apparent difference between cells infected with rH9N2 virus containing either 172E-NS1 or 172K-NS1 (Fig. 2c). These results suggest that the biological impact of G540A substitution in the NS segment may not lie in the amino acid change at position 172 of the NS1 protein.

As position 540 is located within exon 2 of NEP and close to the splice acceptor site, we reasoned that this substitution may affect NEP mRNA splicing. Using ESEfinder to screen the sequence[33], we identified a potential exonic enhancer site (ESE), which spans nucleotides 535–541 of the NS segment. Notably, WT H7N9 NS (540A) has a score of 2.64, whereas the A540G mutant showed a much higher score of 5.22, indicating that the mutant is more efficiently spliced (Fig. 1a). To verify that the G540A substitution affects ESE function in NEP mRNA splicing regulation, we compared NEP/NS1 mRNA ratios in WT and A540G mutant virus-infected cells using primer sets to specifically amplify NEP and NS1 mRNAs (Fig. 2d). As expected from the higher ESE score, A540G increased the proportion of the spliced form of NS mRNA, NEP mRNA, in infected cells, starting from 6 h post infection (Fig. 2e). To exclude the effect of other viral proteins on regulation of NS mRNA splicing, transfection experiments with WT and mutated NS segments were performed and the result showed that A540G back mutation significantly increases the NEP/NS1 mRNA ratio in HEK293T cells (Supplementary Fig. 3a). RNA stability testing showed no differences between WT and A540G mutant mRNAs (Supplementary Fig. 3b). Notably, the effect of A540G back mutation on NEP/NS1 mRNA ratios is not apparent in avian DF-1 cells infected with rH9N2 virus (Supplementary Fig. 3c). Avian cells (DF1) express very low levels of SF2 compared with

mammalian cells (Supplementary Fig. 3d,e) and it is therefore possible that avian cells are less sensitive to splicing changes caused by A540G substitution in the NS RNA. To verify the contention that the effect of G540A is due to the nucleotide substitution, rather than the change of amino acid at position 172, stop codons were introduced at different positions within the NS1 open reading frame to interrupt protein translation from the plasmid. There are several in-frame methionine residues within the NS1 open reading frame; NS-45-stop would only make truncated NS1 with the first 26 amino acids from the amino terminal, whereas NS-456-stop and NS-501-stop would abort translation of NS1 before residue 172 (Fig. 2f). Using these plasmids in a transfection experiment, we further examined NEP/NS1 mRNA ratios. Consistent with the virus infection result, comparison of splicing ratios for NS mRNAs transcribed from transfected NS plasmids encoding WT or truncated NS1 proteins showed that A540G substitution increased the splicing ratio of NS mRNAs (Fig. 2g). To confirm that the NEP-ESE motif identified here is functional in other settings, we conducted a mini-gene assay in which the NEP-ESE motif was inserted into the SMN mini gene to replace the original ESE motif in exon 7 (Fig. 2h)[34]. The NEP-ESE motif was able to function, as indicated by exon 7 inclusion in SMN1 mini gene mRNA splicing, and furthermore the A540G substitution also enhances the efficiency of exon-7 inclusion (Fig. 2h). Taken together, these results clearly characterize a new ESE located in the 535–541 region of exon 2 of NEP pre-mRNA and demonstrate that an A to G substitution at position 540 significantly affects the ability of this ESE to splice NEP mRNA.

**G540A substitution modulates NEP expression**. To further investigate the effect of this newly identified ESE on regulation of NEP mRNA expression, we examined levels of NS1 and NEP protein during virus infection. Using a pair of rH9N2 viruses, which contain either 540A-NS (WT) or 540G-NS segments, expression of viral proteins was examined over the course of infection. Although no significant difference in NP protein level was observed between WT virus and the A540G mutant, NEP and NS1 protein levels were significantly affected in the early hours of A540G mutant virus infection, up until 12 h post infection, with NS1 being downregulated and NEP upregulated (Fig. 3a,b). Interestingly, M1 protein expression was also affected by NS-A540G mutation, presumably due to the altered level of NS1 as reported in other studies, including one in which

**Figure 2 | A540G enhances NS splicing ratio.** (**a**) HEK293T cells were transfected with RNP complexes (PB2, PB1, PA and NP) and vector, different amounts of H7N9-NS1-WT or H7N9-NS1-A540G (K172E) mutant NS1 plasmids, together with firefly (pYH-Luci for RNP activity) and Renilla (control) luciferase reporters, for 16 h. (**b**) HEK293T cells were transfected with vector, different amounts of H7N9-NS1-WT or H7N9-NS1-A540G (K172E) NS1, together with IFN-β and Renilla reporters. At 8 h post transfection, cells were infected or mock-infected with SeV and luciferase activities were analysed after cultured overnight. (**c**) IFN-β mRNA induction in A549 cells mock infected or infected with rH9N2-WT or rH9N2-NS-A540G viruses (multiplicity of infection (MOI) = 1) was analysed by RT–qPCR. (**d**) The black arrowheads indicate primers for detection of NS1 and NEP mRNA used in **e**. (**e**) RNA from A549 cells infected with rH9N2-WT or rH9N2-NS-A540G viruses (MOI = 1) was analysed by RT–qPCR. The splicing ratio was calculated by the ΔCt method. (**f**) Potential in-frame start codons in the H7N9 NS1 gene are shown as grey triangles. The three plasmids used in **g** include different stop codons (red squares), NS-45-stop aims to truncate the NS1 protein in N-terminal and for NS-456-stop and NS-501-stop, to abort translation before amino acid 172 (nucleotide 540). (**g**) HEK293T cells were transfected with pHI-H7N9-NS or its mutants, as indicated, together with H7N9 RNP complexes. The splicing ratio of NS mRNAs was analysed by RT–qPCR. Expression of WT and mutant NS1 proteins was analysed by immunoblotting. The solid arrowhead indicates proteins translated from the second in-frame start codon, while the unfilled arrowhead indicates truncated NS1 proteins ending before residue 172. (**h**) Schematic illustration of SMN mini-gene splicing assay. The verified ESE motif in exon-7 is shown in blue. SMN1 is a wild-type ESE, whereas SMN2 is dysfunctional. The SF2-ESE motif score (ESEfinder) is shown under the sequence. If the ESE is functional, exon 7 would be included during alternative splicing (full length, FL). Otherwise, it would be excluded (−7), leading to a shorter spliced product. RNAs from HEK293T cells transfected with pSMN plasmids containing the indicated ESE motifs were analysed. The ratio of FL to −7 mRNA from the SMN gene was estimated by RT–qPCR. Error bars represent mean ± s.d. (n = 3). Statistical significance was analysed by (**e,c**) two-way analysis of variance with Bonferroni post test or by (**a,b,g,h**) Student's t-test: *P < 0.05, **P < 0.01, ***P < 0.001 and ****P < 0.0001; NS, not significant. Symbols above graph bars or plotted points indicate the statistical significance of the comparison between WT and A540G groups.

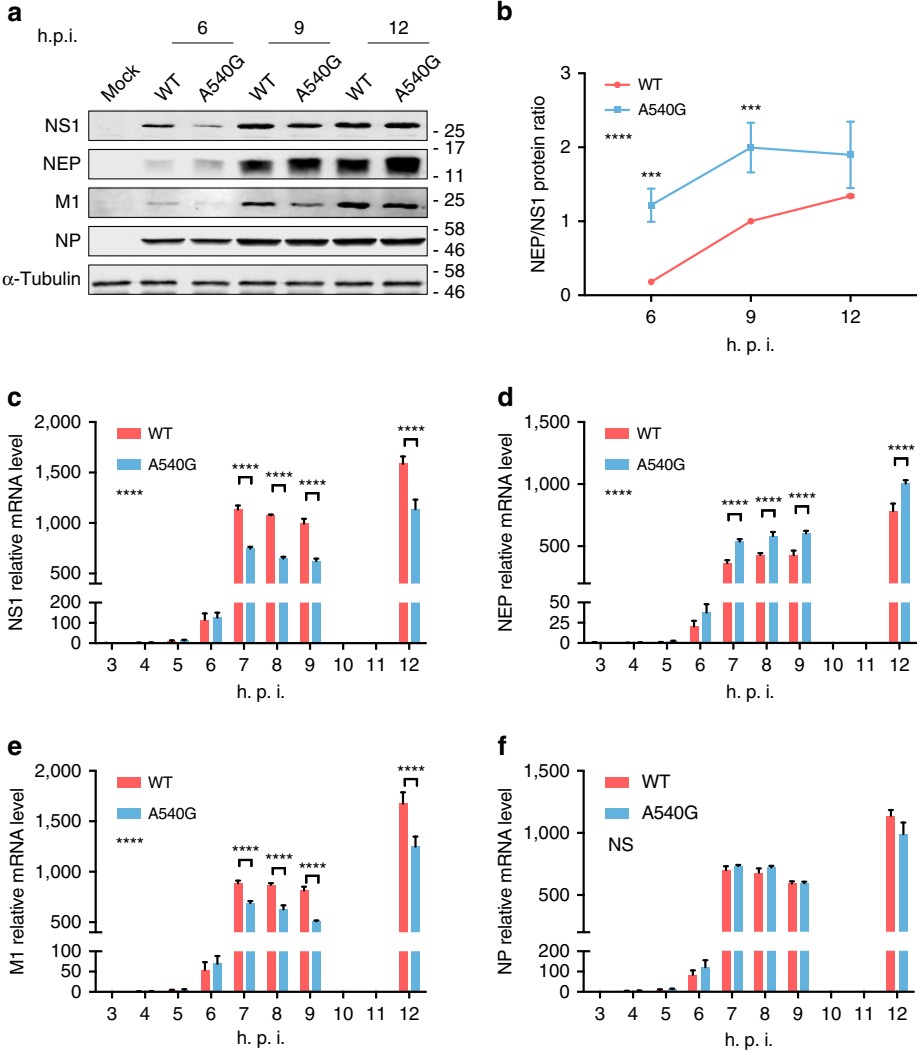

**Figure 3 | Effect of A540G substitution on viral protein expression and mRNA transcription.** (**a**) Whole-cell lysates of A549 cells infected with the indicated H9N2 viruses (multiplicity of infection (MOI) = 1) were analysed by immunoblotting with antibodies against NS1, NEP, NP, M1 and α-tubulin. Expression of viral proteins at different time points is shown. (**b**) The ratio of NEP to NS1 proteins at different time points was calculated using quantification data obtained using Image-J software (**a**). Band intensity of NS1 or NEP are relative to those of WT at 9 h.p.i. Error bars represent mean ± s.d. (n = 3). (**c–f**) Total RNA from A549 cells infected with the indicated viruses at an MOI of 1 was isolated. NS1 (**c**), NEP (**d**), M1 (**e**) and NP (**f**) mRNAs were measured by RT–qPCR and normalized to β-tubulin levels. Error bars represent mean ± s.d. (n = 3). Statistical significance was analysed by two-way analysis of variance with Bonferroni post test: ***$P < 0.001$ and ****$P < 0.0001$; NS, not significant. Symbols on the left of the graphs indicate the statistical significance of the comparison between WT and A540G groups. Full-size western blottings are provided in Supplementary Fig. 8.

NS1 protein was found to promote M1 to M2 splicing in nuclear speckles[9,35]. To further confirm the notion that the NEP ESE modulates NEP and NS1 expression during virus infection, we examined levels of viral mRNAs in A549 cells infected with either WT or A540G-NS rH9N2 virus. Consistent with the western blot result, quantitative reverse-transcription–PCR (RT–qPCR) showed that in comparison with WT virus-infected cells, unspliced NS mRNA (NS1) is decreased in A540G mutant infections, whereas spliced NS mRNA (NEP) is increased (Fig. 3c,d). Similarly, unspliced M1 mRNA levels are down-regulated when NS1 expression is decreased in A540G mutant virus infected cells (Fig. 3e). In contrast, mutation of this ESE did not affect NP mRNA levels during the early hours of virus infection (Fig. 3f). These results further demonstrate that the NEP ESE is associated with splicing of NS mRNAs, and that the G540A substitution in this ESE affects the ratio of NS1 to NEP mRNAs, leading to more NEP protein being expressed in the early hours of infection.

**SF2 binds NEP exon 2 ESE site**. A motif search using ESEfinder indicates that the NEP ESE site is similar to the consensus SF2/ASF-binding sites[33]. Reports indicate that human splicing factor SF2/ASF specifically recognizes the pre-mRNA 5′-splice site[36–38]. To test whether the NEP ESE site is recognized by SF2, we conducted an electrophoretic mobility shift assay (EMSA) using RNA probes derived from different regions of NS mRNA (Fig. 4a). We showed that full-length FLAG-tagged SF2 in nuclear extract (NE) prepared from plasmid transfected HEK293T cells is able to bind to a probe derived from nucleotides 529–548, which cover the NEP-ESE site; the specificity of the shifted complex is further verified by binding of anti-FLAG, but not the IgG antibody control, to tagged SF2 (Fig. 4b). SF2 is an RNA-binding protein composed of an N-terminal RNA recognition motif and a carboxy-terminal RS domain, which is rich in arginine/serine repeats and interacts with other splicing regulators. To test the direct binding of SF2 to NS mRNA, we constructed a FLAG-tagged-SF2 del-RS mutant, which only

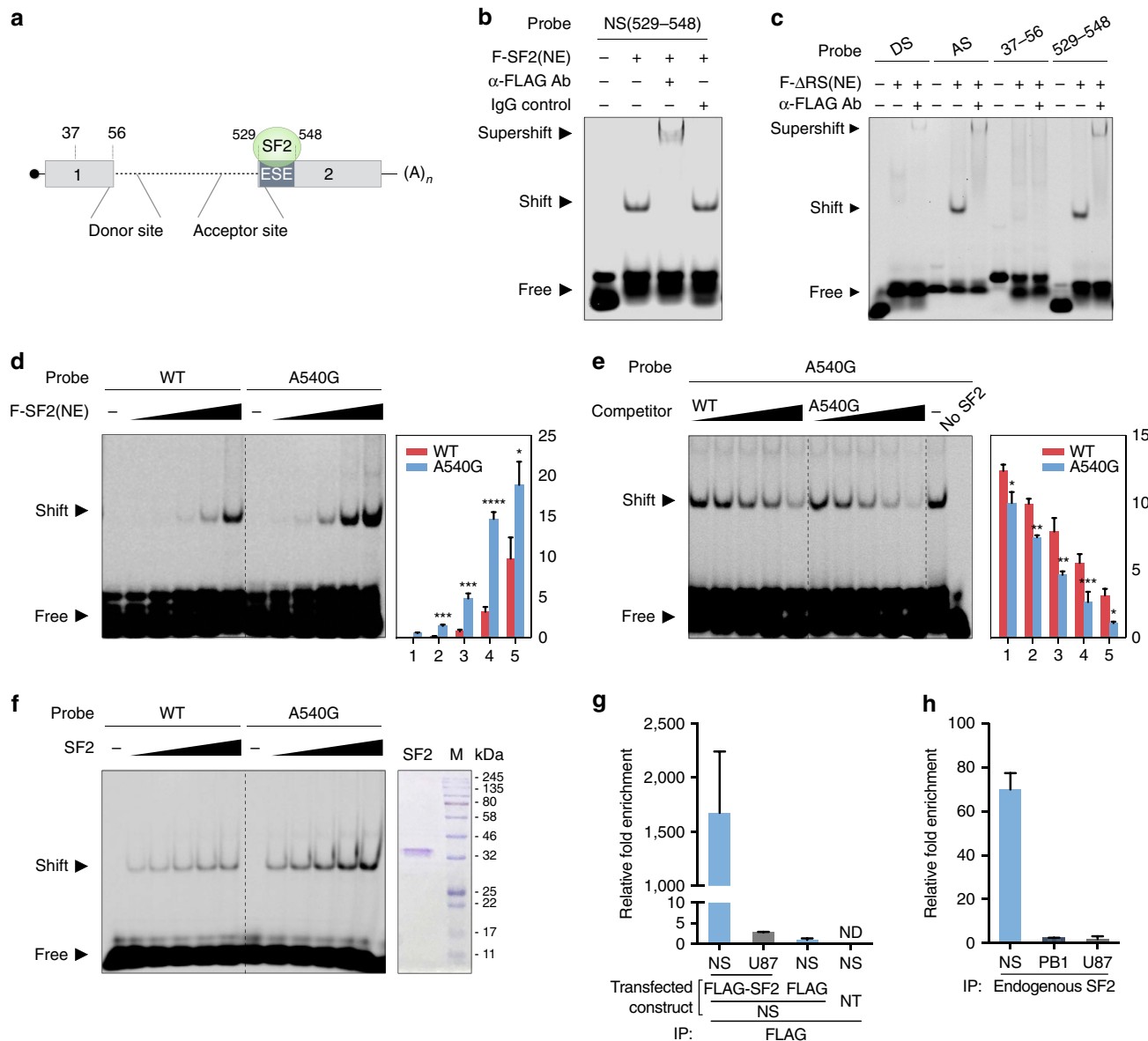

**Figure 4 | SF2/ASF interacts with NS mRNA through multiple binding sites including the ESE motif and A540G substitution enhances SF2 binding to NEP ESE.** (**a**) Schematic illustration of the SF2/ASF-binding sites in NS mRNA and locations of probes used in the following experiments. (**b**) SF2/ASF binds to NS (529–548), in which the putative ESE is located, in RNA EMSA using NE containing FLAG-tagged SF2/ASF. α-FLAG antibody was used for super-shifting and IgG was used as a control antibody. (**c**) Different regions of NS mRNA bound by SF2/ASF were analysed by RNA EMSA using FLAG-SF2ΔRS. RNA probes derived from the splicing donor site (DS), acceptor site (AS), exon-1 (37–56) and exon-2 (529–548) of NS mRNA were similarly tested, as in **b**. (**d**) SF2 binding efficiency to WT and A540G mutant ESE was compared in RNA EMSA with increasing amounts of FLAG-SF2 NE. (**e**) Competition efficiency with WT or A540G cold probes was compared in RNA EMSA with increasing amounts of cold probes (competitors). Results displayed in **d**,**e** are representative of three separate experiments. Error bars represent mean ± s.d. ($n = 3$). Statistical significance was analysed by Student's t-test: *$P < 0.05$, **$P < 0.01$, ***$P < 0.001$ and ****$P < 0.0001$. (**f**) SF2 was purified by IP in high-salt RIPA buffer using α-FLAG antibodies and the purity of FLAG-SF2 verified by Coomassie blue staining. Efficiency of SF2 binding to WT and A540G mutant ESEs was compared in RNA EMSA with increasing amounts of purified FLAG-SF2. The relative intensity of the shifted bands was quantified using Image-J and is shown at the right side of each graph (**d**,**e**). (**g**) RNA-IP assay with lysates prepared from pHW2000-H7N9-NS transfected cells co-transfected with FLAG-SF2, FLAG-vector or mock transfected (NT). Whole-cell lysates of HEK293T cells were prepared at 48 h post transfection and subjected to IP with α-FLAG. (**h**) RNA-IP assay in virus infected cells. Whole-cell lysates were prepared from A549 cells infected with rH9N2-WT virus (multiplicity of infection (MOI) = 2) at 16 h post infection and subjected to IP with α-SF2/ASF (endogenous). (**g**,**h**) RNA-IPs were followed by RNA extraction and RT–qPCR with primers detecting viral NS, cellular U87 scaRNA or viral PB1 mRNA. Fold enrichment of mRNA was calculated by the ΔCt method. The error bars represent mean ± s.d. ($n = 3$).

retains the RNA-binding motif[39]. We confirmed that the ESE in exon 2 of NEP mRNA is recognized and bound strongly by SF2 del-RS, using both the acceptor site and NEP-ESE (529–548) probes (Fig. 4c). In contrast, a probe derived from nucleotides 37–56 in NEP exon 1 is not bound by SF2 del-RS, whereas very

weak binding was detected between the NEP donor site and SF2–RS (Fig. 4c). As NEP ESE-540A and ESE-540G exhibit different levels of NEP mRNA splicing efficiency, we tested whether SF2 has different binding affinity for the 540A and 540G variants. EMSA showed that SF2 has higher affinity for the NEP

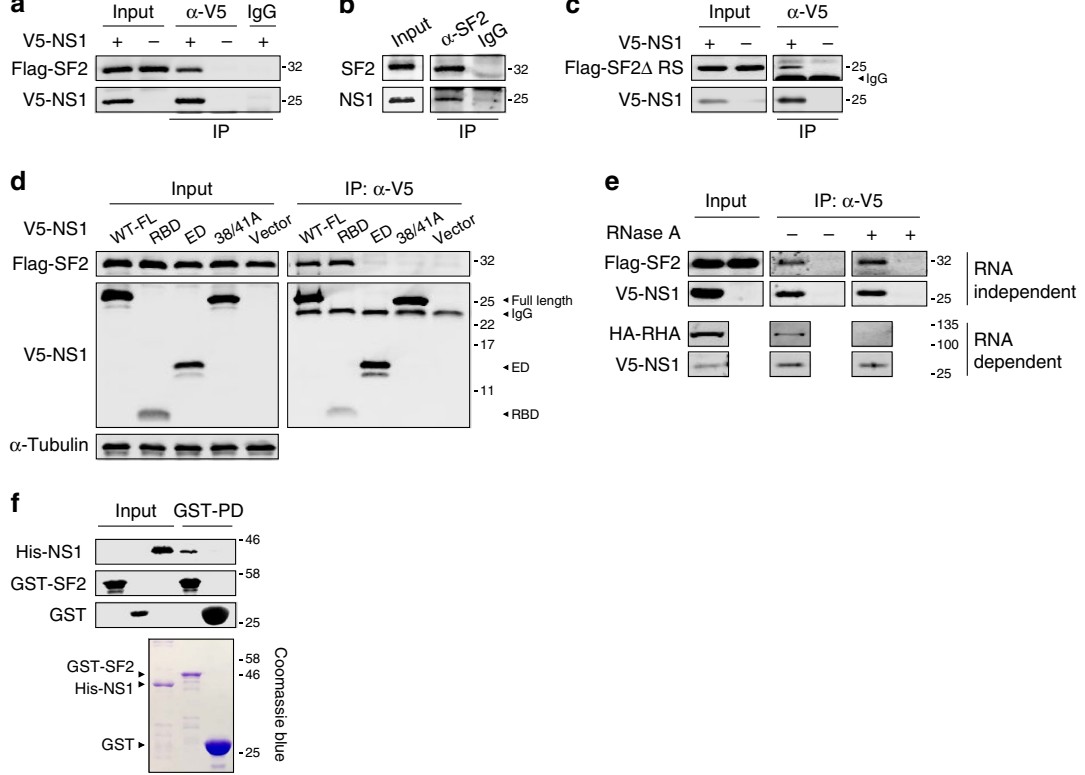

**Figure 5 | NS1 interacts with SF2/ASF.** (**a**) HEK293T cells were co-transfected with FLAG-SF2 and V5-NS1 or V5-vector, and cell lysates used for IP with α-V5 or control IgG, followed by immunoblotting (IB) with α-FLAG and α-V5. (**b**) A549 cells were infected with rH9N2-WT viruses at an multiplicity of infection (MOI) of 5 for 16 h and cell lysates used for IP with α-SF2/ASF (endogenous) or control IgG, followed by IB with α-SF2/ASF and α-NS1. (**c**) HEK293T cells were co-transfected with V5-NS1 or V5-vector, along with FLAG-SF2ΔRS plasmids and cell lysates were then used for IP with α-V5 or control IgG, followed by IB with α-FLAG and α-V5. (**d**) HEK293T cells were transfected with FLAG-SF2, together with V5-NS1, V5-RBD of NS1, V5-ED of NS1, V5-tagged mutant NS1-R38A/K41A (38A/41A) or V5-vector, and cell lysates were used for IP with α-V5 or control IgG (control IgG blots not shown), followed by IB with α-FLAG and α-V5. (**e**) HEK293T cells were co-transfected with FLAG-SF2 and V5-NS1, with cell lysates being immunoprecipitated with α-V5 or control IgG and washed, then treated or mock-treated with RNase A at 37 °C for 45 min and further washed, followed by immunoblotting with antibodies against FLAG or V5. Co-transfection of HA-RHA (RNA Helicase A) and V5-NS1 is used as a control to demonstrate an RNA-dependent interaction. (**f**) Bacterially expressed GST-SF2, GST (control) and dual His-tagged NS1 proteins were purified and used in a GST pull-down assay, followed by IB with α-His and α-GST. A Coomassie blue-stained SDS–PAGE gel shows the purity of recombinant proteins. Full-size western blottings are provided in Supplementary Fig. 8.

A540G mutant than WT (540A) in both direct and competition binding assays (Fig. 4d,e). These binding experiments clearly demonstrated that the NEP-ESE site (nucleotides 529–548) is recognized by the SF2 splicing regulator, and that A540G substitution increases SF2 binding in *in vitro* assays. To exclude the possible influence of other cellular proteins present in the NEs prepared from SF2-transfected cells, we confirmed that antibody purified SF2 is able to bind to the NEP-ESE site, and that A540G mutant appears to have higher affinity for SF2 (Fig. 4f). To confirm whether this SF2-binding ESE is common among other influenza viruses, a panel of representative strains of different influenza A virus subtypes was screened, with all found to contain an ESE in exon 2 of NEP that could be bound by SF2 (Supplementary Fig. 4a).

We further tested whether SF2 can bind to NS mRNA in cells. RNA (IP) and quantitative RT–PCR assays showed that, compared with precipitation with control antibodies, NS mRNA, but not control cellular RNA (U87), can be enriched more than 80-fold by precipitation of SF2 in HEK293T cells where FLAG-tagged SF2 and the H7N9-NS segment are co-expressed (Fig. 4g). We then used anti-SF2 antibody to precipitate endogenous SF/ASF and confirmed that NS mRNA, but not control cellular or viral RNA (PB1), is efficiently co-precipitated

from H9N2 infected A549 cells (Fig. 4h). Direct interaction between SF2 and NS mRNA was further confirmed by examining co-localization in the nucleus of A549 cells infected with rH9N2 virus (Supplementary Fig. 4b). The EMSA and RNA IP/RT–qPCR assays clearly demonstrate that the NEP-ESE site is directly bound by the splicing regulator SF2, and that A/G substitution at position 540 of the NS segment alters SF2-binding affinity for this ESE site, leading to different splicing efficiency.

**NS1 protein interacts with SF2 in the nucleus.** SF2 is involved in the regulation of splicing of mRNAs transcribed from the M segment[1]. NS1 inhibits host antiviral responses through multiple mechanisms, including suppression of the host mRNA splicing process, and previous studies, including our own, have also shown that viral NS1 protein contributes to the regulation of M2 mRNA splicing[9,18]. However, it is less clear how NS1/NEP mRNA splicing is regulated. We wished to examine whether the NS1 protein interacts with SF2, as a part of the splicing process, during influenza virus infection. Using V5-tagged NS1 derived from H7N9 virus and FLAG-tagged SF2 plasmids in a co-expression assay, and rH9N2-infected A549 cells, we

showed that SF2 and NS1 are co-precipitated from cell lysates (Fig. 5a,b). Consistent with the above result showing that SF2 uses the RNA recognition motif domain to bind to the NEP-ESE site, we demonstrated that the interaction between NS1 and SF2 is independent of the RS domain of SF2 (Fig. 5c). Further characterization of the interaction between NS1 and SF2 found that the RNA-binding domain (RBD) of the NS1 protein is required for interaction with SF2, and that mutations at positions 38 and 41 abolish this interaction (Fig. 5d). Although positions 38 and 41 are both important for NS1 RNA binding[40], our result showed that only R38A/K41A double mutations abolished interaction between NS1 and SF2 (Supplementary Fig. 5a), suggesting double mutations at positions 38 and 41 may alter the conformation of the NS1 RBD, affecting interaction with SF2. It is noted that the F162S/V163R mutant, which reduces SF2 RNA binding ability, did not affect SF2 interaction with NS1 (Supplementary Fig. 5a,b). To further exclude the possibility that RNA mediates the interaction between SF2 and NS1, RNAs were removed by treatment with RNase A and SF2 was still seen to interact with NS1 (Fig. 5e). In contrast, the RNA-dependent interaction between the NS1 protein and RNA Helicase A is abolished with RNase A treatment[41]. To verify that the interaction between SF2 and the NS1 protein is direct, and does not involve other cellular proteins, purified SF2 and NS1 proteins were used to confirm SF2–NS1 direct interaction in a glutathione S-transferase (GST) pull-down assay (Fig. 5f). Finally, bimolecular fluorescence complementation (BiFC) images showed association of NS1 and SF2 in transfected cells and immunostaining of SF2 and NS1 proteins in the early hours of virus infection showed spots of NS1 and SF2 co-localization in the nucleus, further confirming interaction of these proteins (Supplementary Fig. 5c,d). These results clearly demonstrate a direct interaction between SF2 and the NS1 protein during influenza virus replication.

**NS1 modulates expression of NEP through interaction with SF2.** The NS1 protein regulates the host splicing process by interacting with nuclear small RNA to prevent the formation of spliceosome complexes and blocks nucleocytoplasmic transport of host mRNAs to intervene with expression of genes required for host antiviral functions[17,42,43]. NS1 was shown to modulate the splicing of M2 mRNA in virus replication[9,18,44]. However, the question of whether NS1 is also involved in regulating splicing of NS mRNA into NEP remains unresolved. As the NEP exon 2 ESE motif is bound by SF2 and NS1 protein interacts with SF2, we attempted to re-visit the question of whether NS1 is involved in the process that produces spliced NEP mRNA during virus replication. To this end, we constructed an NS-null replicon system to estimate NS mRNA splicing without expressing either NS1 or NEP protein. In this system, NS vRNA is first generated by a polI-driven plasmid and then NS mRNA is transcribed from vRNA by viral polymerase (Fig. 6a). The effect of NS1 is examined by inclusion of plasmids expressing different versions of NS1, driven by the regular pol II promoter. Using this NS-null replicon system, we found that expression of the NS1 protein reduces the NEP/NS1 ratio in a dose-dependent manner (Fig. 6b). Further analysis revealed that while both the RBD and effector domain decrease the NEP/NS1 mRNA ratio, full-length NS1 is required for maximum inhibitory activity (Fig. 6c). A previous study demonstrated that a 148A/152A/153A NS1 mutant facilitated export of NS1 to the cytoplasm and prevented its nuclear accumulation (Supplementary Fig. 6)[45]. We found this mutant to be significantly attenuated in its ability to inhibit NEP mRNA splicing (Fig. 6d), suggesting the inhibitory action of NS1 occurs in the nucleus. Consistent with the SF2 interaction result, we found that the NS1 mutant, which is unable to interact with

SF2 (R38A/K41A), is also attenuated in its ability to inhibit NEP mRNA splicing even though it is localized in the nucleus (Fig. 6e and Supplementary Fig. 6), implying that the role of NS1 in regulation of NEP splicing is mediated by SF2. It is possible that SF2 mediates NS1 inhibition of the NEP mRNA splicing process. To test this possibility, we knocked down SF2 in HEK293T cells and tested the effect of NS1 on the NEP/NS1 mRNA ratio. Using a NS-null replicon system, we found that restoring NS1 expression has a less inhibitory effect on splicing of NEP mRNA in cells where SF2 is efficiently knocked down, as seen by a significantly increased ratio of NEP/NS1 mRNA (Supplementary Fig. 6b). Taken together, these results suggest that expression of NS1 has an inhibitory effect on splicing of NEP mRNA and that this process, at least partly, involves interaction with SF2 in the nucleus.

**G540A substitution enhances virus virulence in mice.** An A540G mutation within the identified NEP-ESE site affects H7N9 virus replication in mammalian cells, but not in those of avian origin (Fig. 1b). To characterize the effect of WT and A540G mutant viruses *in vivo*, we compared the pathogenic properties of H7N9 viruses containing either 540 A-NS (WT) or 540G-NS (mutant) in a mouse model. BALB/c mice were infected with different doses of H7N9 virus and observed for mortality and morbidity. Although all infected mice survived the low inoculum dose ($4.75 \times 10^3$ plaque-forming units) of either strain of H7N9 virus (Fig. 7b), substitution of A540G significantly attenuated virus pathogenicity, as shown by the body weight curve (Fig. 7a). At higher infectious doses ($4.75 \times 10^4$ and $4.75 \times 10^5$ plaque-forming units), both strains caused severe body weight loss in mice, but the A540G-NS mutant was still less pathogenic than the WT virus. Estimation of $MLD_{50}$ clearly demonstrated that WT virus is three times more pathogenic in mice. However, there was no significant difference between the WT and A540G groups when virus titres in lung tissues collected 3 days after infection were measured, implying that 540A may associate with virulence properties other than virus replication ability *in vivo* (Supplementary Fig. 7). We then analysed mRNA levels of cytokines in the lungs of these mice, but found no significant difference between WT and A540G virus-infected mice. As cell culture experiments showed that A540G substitution has a more significant effect on virus protein expression during the early stage of virus infection (Fig. 3a), it is possible that day 3 post infection may be too late to detect any differences in virus replication *in vivo*. These results confirm that position 540 within the NEP ESE in the NS segment is associated with virus pathogenicity in mammalian cells *in vivo*, but the mechanism underlying the enhanced virulence of genotype A540-NS H7N9 virus needs to be further investigated.

**Discussion**
Cross-species transmission of influenza virus is normally blocked by host restriction factors, which limit virus infection and replication in new hosts. Although influenza virus transcribes and replicates the viral genome using the viral polymerase complex, it relies on host splicing machinery for processing of viral mRNAs. In addition to adaptive mutations of viral functions such as the polymerase complex, regulation of M and NS expression through modulation of mRNA splicing is implicated in influenza virus replication efficiency[19,20,46]. This study identified an ESE located in exon 2 of NEP in the NS segment of the H7N9 virus, and found it to be common and relatively conserved among other influenza A virus subtypes. Notably, a unique G540A substitution within this ESE was found in H7N9 and also in the H9N2 virus, which provided internal genes to the H7N9 virus. G540A

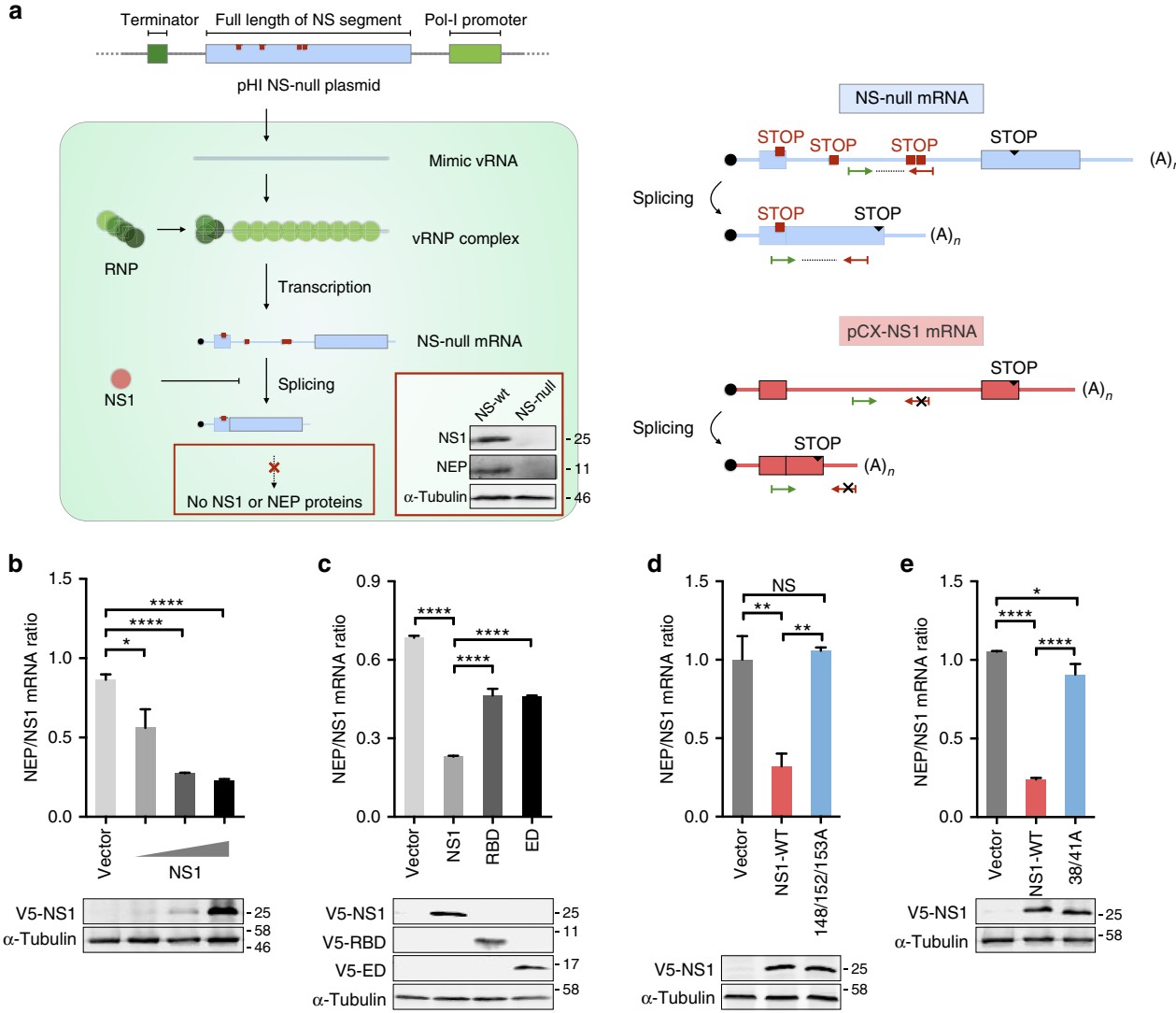

**Figure 6 | Effect of NS1 on NS mRNA splicing. (a)** Schematic illustration of NS-null replicon system and detection of specific NS mRNAs. (Left panel) Construction of the pHI NS-null plasmid, which contains multiple stop codons (red squares). NS vRNA mimics were generated from pHI NS-null plasmid by pol-I and incorporated into RNP complexes. NS vRNA mimics were further transcribed into NS-null spliced and unspliced mRNAs by the viral polymerase complex and host machinery. Owing to the presence of stop codons, neither unspliced or spliced NS-null mRNAs could be translated into complete NS1 or NEP proteins, respectively. (Right panel) Specific primers (arrowheads) for detection of NS-null mRNA and its spliced form, but not mRNA expressed from the pCX-NS1 plasmid, in RT–qPCR assay. The primer sets only detect NS-null mRNA and not NS1 mRNA expressed from the pCX-NS1 plasmid, as the reverse primer (red) is base pair matched to the introduced stop codons and not WT sequences. Meanwhile, the antisense primer for spliced NS-null mRNA anneals to the coding region of NEP that is not present in NS1 mRNA. Validation of NS1 or NEP protein expression when pHI-H7N9-NS or pHI-H7N9-NS-null are co-transfected together with the RNP polymerase complex, is shown by IB using antibodies against NS1, NEP or α-tubulin (insert, left panel). **(b)** HEK293T cells were transfected with pHI-NS-null and RNP, together with increasing amounts of V5-NS1 (0.5, 5 and 50 ng). **(c–e)** HEK293T cells were transfected with pHI-NS-null and RNP, together with either vector (control) or WT, RBD or effector domain (ED) versions of V5-NS1 (**c**), WT or 148/152/153A mutant of V5-NS1 (**d**), or WT or R38A/K41A V5-NS1 (**e**). Total RNA from transfected cells in **b,c,d** and **e** was isolated and the splicing ratio of NS-null mRNA was measured by RT–qPCR. Levels of the various versions of NS1 and tubulin (control) were analysed by IB with antibodies against V5 and tubulin, respectively. Error bars represent mean ± s.d. ($n = 3$). Statistical significance was analysed by Student's t-test: *$P < 0.05$, **$P < 0.01$ and ****$P < 0.0001$.

enhances virus replication in mammalian cells, while retaining replication ability in avian cells. Furthermore, the NEP-ESE interacts with splicing regulator SF2 and viral NS1 protein interacts with SF2 to modulate NEP/NS1 mRNA splicing in the nucleus during virus infection. It seems likely to be that an adaptive mutation creating an optimal NEP-ESE may contribute to the ability of the H7N9 virus to replicate in humans, while retaining fitness for circulation in poultry.

The M2 and NEP proteins are expressed from the M and NS segments, respectively, through a splicing process after transcription. The splicing of influenza virus mRNAs relies on host machinery and is highly regulated. Regulation of M2 mRNA is mediated by binding of the splicing regulator, SF2, to the splicing enhancer located in exon 2 of M2 mRNA[1]. It has been suggested that the viral polymerase complex is involved in regulation of M2 mRNA splicing regulation through interaction with the splice site in exon 2 and other studies have also argued that viral polymerases have a role in this process[44,47,48]. Evidence from our own and other studies have suggested a mechanism involving NS1 in the regulation of splicing of

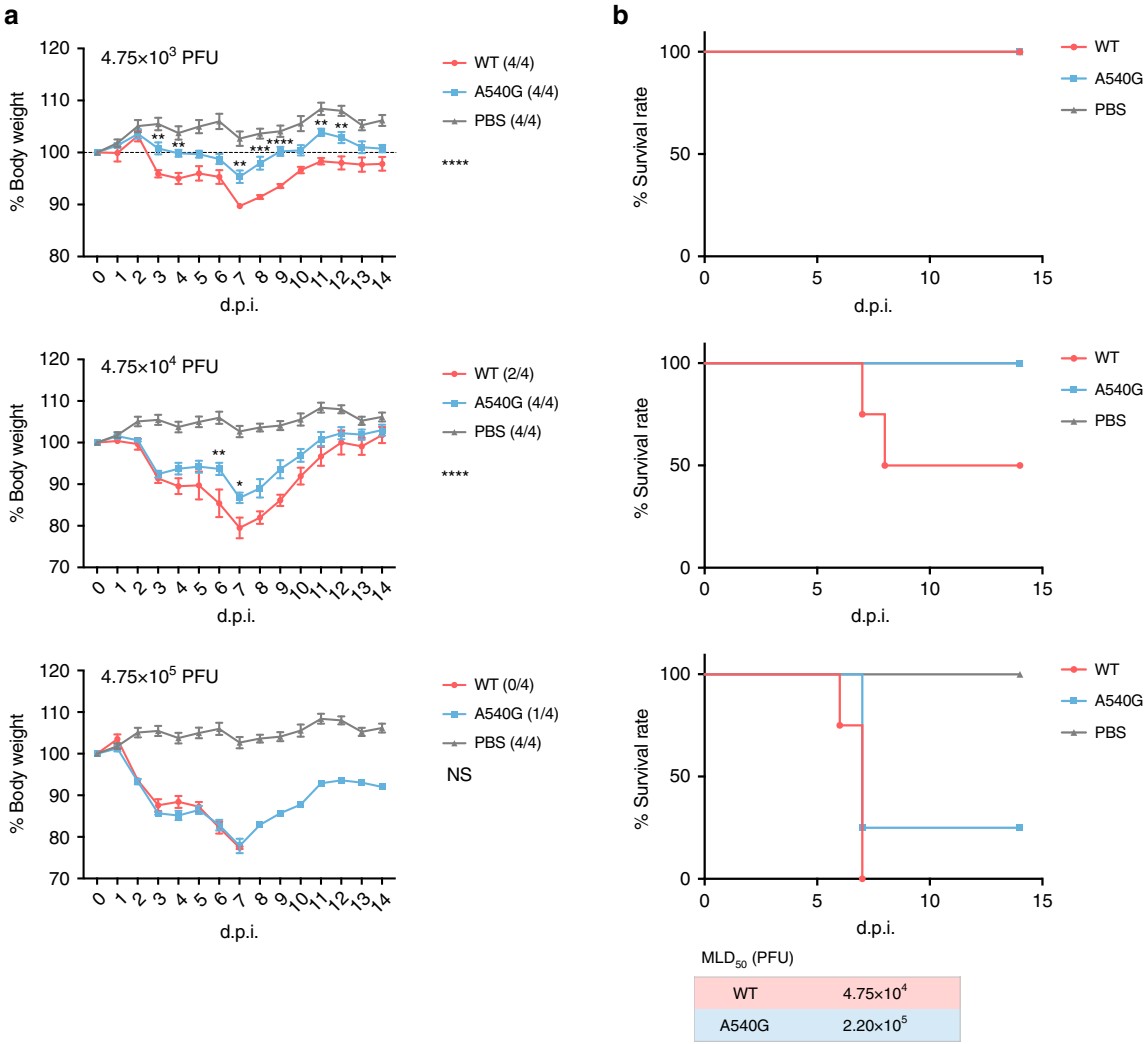

**Figure 7 | Pathogenicity of H7N9-WT and H7N9-NS-A540G viruses in mice.** Six- to 8-week-old female BALB/c mice were intranasally inoculated with the indicated doses of H7N9-WT or H7N9-NS-A540G viruses, or PBS control. (**a**) Weight loss and (**b**) mortality were monitored daily for 2 weeks. Numbers in brackets in graph keys show the final number of surviving mice. $MLD_{50}$ was calculated by the method described previously[59]. Body weight differences between the WT H7N9 and H7N9-NS-A540G groups were compared and statistically analysed. Error bars represent mean ± s.e.m. ($n = 4$). Statistical significance was analysed by two-way analysis of variance with Bonferroni post test: *$P < 0.05$, **$P < 0.01$ and ****$P < 0.0001$; NS, not significant. d.p.i., days post infection. Symbols on the right of the graphs indicate the statistical significance of the comparison between WT and A540G groups.

M mRNAs, as the M2/M1 mRNA ratio is altered in viruses lacking the NS1 protein[9,44,49]. However, it is less clear how expression of the spliced form of NS mRNA is regulated to influence NEP expression. Besides its defined role in mediating nuclear export of the viral RNP complex, a role for NEP in regulating the switch from transcription to replication during virus replication is being investigated[50–52]. It is suggested that NEP is regulated to remain at low levels to avoid negative effects on viral transcription during the early phase of infection[20]. Studies on PB2 host adaption also found that a mammalian PB2 adapts better than avian PB2 for promoting RNP polymerase activity in the presence of NEP in mammalian cells[28,51]. It seems that the level and timing of NEP expression significantly affects virus replication efficiency. However, questions about NS1 protein involvement in the regulation of its own mRNA splicing process remain unanswered due to contradictory reports from studies using different experimental systems[10,17,18]. The results presented here appear to support the hypothesis that NS1 is involved in regulating splicing of NEP mRNA through interaction with SF2. NS1 is one of the earliest proteins

expressed during virus replication and may be the key regulator in determining virus replication efficiency, through modulation of the NS and M splicing processes. As expression of NEP and NS1 is mutually exclusive, relative levels of NEP/NS1 could affect virus replication efficiency and virulence properties. Higher levels of NS1 in the nucleus in the early stages of virus infection may further modulate viral protein expression and inhibit host antiviral gene expression, thereby promoting replication. Coordinated timing of the expression of these viral functions may synergistically support higher viral replication efficiency (Fig. 8). However, a mechanism detailing how an optimized balance of NEP/NS1 is achieved and adjusted over the course of influenza virus infection, to ensure high efficiency of virus replication remains to be elucidated. It is possible that more than one mechanism may be involved in regulation of splicing of NS mRNAs, as another study reported that influenza virus polymerases are able to recruit the host protein complex, RED-SMU1, to regulate splicing of NS mRNAs[2].

The endemicity of the H7N9 virus, along with the persistent reemergence of human infections caused by this virus, is

**Figure 8 | Working model for the regulation of NS mRNA splicing.** NEP mRNA is a spliced form of NS1 mRNA. The ESE, a *cis*-acting element located in the second exon, positively regulates the alternative splicing of NS1 mRNA. SF2/ASF binds to the newly identified ESE site and facilitates the splicing process. NS1 protein shuttles between the cytoplasm and the nucleus, and suppresses the splicing of NS1 mRNA through binding to SF2. NS1 has multiple functions for suppression of host antiviral activities in the nucleus and cytoplasm, whereas NEP is involved in vRNP export to the cytoplasm for virion assembly and packaging, and may also be involved in directing the switch from viral genome transcription to replication. As the production of NEP and NS1 mRNA is mutually exclusive, regulation of splicing of NS mRNA has an impact on viral replication efficiency.

unusual[53,54]. This study provides a possible mechanism to explain the molecular properties, which allow H7N9 virus to infect humans while retaining the ability to circulate in avian species. Current NS-540A genotype H9N2 and H7N9 viruses replicate better in mammalian cells than mutant versions with a back mutation to A540G, but retain their ability to replicate in avian cells. The primed condition of NS-540A for enhanced replication ability in mammalian hosts may facilitate viral acquisition of other adaptive mutations, such as PB2 627K, during virus infection. It is possible that mutations in other gene segments may also have contributed to the emergence and predominance of the NS-G540A H9N2 virus, which provided the internal genes for the H7N9 virus[55], and it is suggested that antigenic variations may also have facilitated the emergence of an H9N2 variant in China[29]. It is notable that human infections with H10N8 confirmed in China were caused only by H10N8 viruses, which contain NS-G540A; no human cases involving H10N8 virus without this mutation were found[24,56]. Human infections with H5N6 virus occurred in China since 2014, with 11/16 (69%) of isolates characterized from human cases contain NS-G540A[57]. The mechanism underlying how this mutation in the here characterized NEP-ESE supports virus replication in mammalian cells should be further evaluated and future surveillance should also monitor for adaptive substitutions, which affect splicing of viral mRNAs.

## Methods

**Viruses and reverse genetics.** All the influenza A viruses used in this study were rescued by reverse genetics technique[58]. For H7N9-WT virus, all eight segments were derived from the strain A/Zhejiang/DTID-ZJU01/2013. For reassortant rH9N2-WT or rH9N2-NS-A540G viruses, HA and NA surface genes were derived from A/HongKong/308/2014 (H9N2), whereas internal genes were identical to H7N9-WT or H7N9-NS-A540G. A pair of H9N2 viruses containing WT or NS-A540G were made with the genome from the strain A/HongKong/308/2014. Viral genotypes were confirmed by sequencing viral genomes.

**Plasmid construction.** Expression plasmids pCX-V5-H7N9-NS1 and pcDNA-FLAG-SF2 were constructed by ligation independent cloning using exonuclease III as previously described[28]. *SMN* mini gene was amplified as described[34], cloned

into pcDNA3.1 vector and designated pSMN. The vector pHI was derived from pHW2000 plasmid by deleting the pol-II promoter and used for cloning NS-null plasmids. To generate mutant plasmids, site-directed mutagenesis was carried out with *PfuTurbo* DNA polymerase.

**Growth kinetics of viruses in cells.** Confluent human A549 (American Type Culture Collection (ATCC), CCL-185) or chicken DF-1 (ATCC, CRL-12203) cells were infected with indicated viruses at a multiplicity of infection of 0.01. The viral inoculums were discarded after 1 h of virus adsorption. Cells were then washed with PBS and cultured in minimal essential media containing $1 \mu g \, ml^{-1}$ of TPCK-treated trypsin. Culture supernatants were harvested at indicated time points and viral titers were measured by plaque assay.

**Transient transfection and *SMN* mini genes assay.** HEK293T (ATCC, CRL-3216) cells were used for transfection assay using TransLT-1 reagent (Mirus Bio) according to the user manual. Mini-gene expression plasmid pSMN-1, dysfunctional mutant pSMN-2, pSMN-H7N9-WT containing the putative ESE of H7N9 or pSMN-H7N9-A540G was transiently transfected into HEK293T cells for 24 h. Total cellular RNAs were extracted and reversely transcribed into complementary DNA. PCR using Hot Start *Taq* (Takara) was performed to amplify cDNA. PCR products were analysed by agarose gel electrophoresis.

**Dual luciferase reporter assay.** To measure the RNP activity, HEK293T cells were co-transfected with RNP expression plasmids composed of PB2, PB1, PA and NP, together with increasing doses of NS1 expression plasmids, pYH-Luci reporter, a kind gift from Professor Robert Webster St Jude Children Research Hospital), and *Renilla* reporter (Promega) and activity estimated 20 h post transfection. To determine the suppression of IFN-β, HEK293T cells were co-transfected with increasing doses of NS1 expression plasmids, IFN-β reporter and *Renilla* reporter. At 8 h post transfection, cells were infected or mock infected with Sendai virus, a kind gift from Professor Dong-Yan Jin (the University of Hong Kong), overnight before being lysed. Luciferase and *Renilla* luciferase activity were measured according to the manufacturer's manual. The activity of luciferase was normalized to that of *Renilla*.

**Reverse transcription and quantitative PCR.** Total cellular RNAs were isolated using RNAiso reagent (Takara). Elimination of DNA and reverse transcription (RT) of RNA were performed using a PrimeScript RT Reagent Kit with genomic DNA Eraser (Takara), in accordance with the manufacturer's manual. Oligo-dT primer was used in RT reaction for detection of mRNAs, whereas uni-12-specific primer was used for vRNAs. Quantitative PCR (qPCR) mixtures were prepared according to the user manual of SYBR Premix Ex Taq reagent (Takara) and reactions were run in a LightCycler 480 instrument II (Roche). For relative quantification, the ΔCt (threshold cycle) method was used to analyse splicing ratios and the ΔΔCt method was adopted for gene expression analysis, where α-tubulin or Rpl32 mRNA expression levels were used for normalization. Primers for qPCR are listed in Supplementary Table 1.

**Pathogenicity study in mice.** Six- to 8-week-old female BALB/c mice were intranasally inoculated with the indicated doses of H7N9-WT or H7N9-NS-A540G diluted in PBS. Weight and survival were recorded daily for 2 weeks. Mice that lost ≥ 20% of initial weight were killed. The protocols for animal experiments were approved by the CULATR (3653-15). $MLD_{50}$ was calculated by the method described by Reed and Muench[59].

**Immunoprecipitation and western blotting.** For RNA IP, HEK293T cells were co-transfected with pcDNA-FLAG-SF2 and pHW2000-H7N9-NS for 48 h, whereas A549 cells were infected with rH9N2-WT at an multiplicity of infection of 5 for 16 h. Cell lysates were incubated with antibodies (1 μg per reaction in 200 μl of lysis buffer) against FLAG tag (Sigma, F1804), endogenous SF2/ASF (Santa Cruz, sc-33652) or control IgG antibodies (Santa Cruz, sc-2025) at 4 °C overnight and precipitated with Dynabeads (Thermo Fisher Scientific) at room temperature for 15 min. After five washes with Tris lysis buffer, RNAs were isolated and analysed by RT–qPCR. For co-IP, HEK293T cells were co-transfected with pcDNA-FLAG-SF2 (or truncated/mutated forms) and pCX-V5-H7N9-NS1 (or truncated/mutated forms), or V5 vector for 48 h, whereas A549 cells were infected as described above. Cell lysates were incubated with antibody–bead complexes at room temperature for 30 min. Protein–antibody–bead complexes were washed five times with Tris lysis buffer, boiled at 95 °C for 10 min and analysed by western blotting. For co-IP with RNase treatment, HEK293T cells were cotransfected with pCX-V5-H7N9-NS1 and pcDNA-FLAG-SF2 or HA-RHA for 48 h. Cell lysates were incubated with antibody–bead complexes and washed three times, as described above. Protein–antibody–bead complexes were then treated or mock-treated with RNase A (ThermoFisher Scientific, 12091039, 1:1,000) at 37 °C for 50 min. Complexes were washed three more times, boiled at 95 °C for 10 min and analysed by western blotting. Endogenous SF2 protein was detected using rabbit polyclonal

antibody (Abcam ab38017, 1:1,000). Uncropped western blottings are provided in Supplementary Fig. 8.

**GST pull-down assay.** BL21 (DE) competent cells harboring pGEX6P-SF2, pGEX6P-1 or pET32a-dualHis-H7N9-NS1 were amplified and induced by treatment with 0.1 mM isopropyl-β-D-thiogalactoside for 4 h at 30 °C. GST-tagged-SF2 and His-tagged NS1 proteins were purified with Glutathione Sepharose beads (Promega) and Ni-NTA beads (Qiagen), respectively. The purity of recombinant proteins was determined by SDS–PAGE. Five micrograms of recombinant NS1 protein was mixed with GST–SF2–bead or GST–bead (control) complexes at 4 °C overnight. Protein–bead complexes were washed four times with Tris lysis buffer and analysed by western blotting.

**RNA electrophoretic mobility shift assay.** HEK293T cells were transfected with pcDNA-FLAG-SF2 or pcDNA-FLAG-SF2ΔRS. Cells were harvested after 24 h, washed by PBS and lysed with sucrose buffer containing NP-40. Nuclear fraction was spun down and washed by sucrose buffer without NP40. Nucleus pellets were resuspended in low-salt buffer, incubated with equal volume of high salt buffer for 30 min on ice and NEs were prepared by spun at $14,000 g$ for 15 min (detail protocol can be found in http://www.celldeath.de/apometh/emsa.html). To set up the binding reaction, 10 nM of RNA probes (Integrated DNA Technologies) labelled with IRDye-800 (Li-Cor Bioscience) were incubated with increasing doses of NE in 20 μl of buffered mixture (30 mM Tris-HCl, 24 mM KCl, 800 μM $MgCl_2$, 0.008% NP40, 3.3 mM dithiothreitol, 0.25% Tween 20, 4% glycerol, 2 μg yeast transfer RNA, pH 8.0). For the competition assay, increasing doses of competitors were added into the mixture. For super-shift assay, 0.5 μg of antibody against FLAG tag (Sigma, F1804)) was included. After incubation for 30 min at room temperature in the dark, mixtures were separated by 5% non-denaturing gel (0.5 × TBE). To purify SF2 protein, FLAG-tagged-SF2 was overexpressed in HEK293T cells and immunoprecipitated using antibody-bead complexes specific for the FLAG tag in high-salt RIPA buffer, then eluted with 3 × Flag peptide. Gels were directly scanned by the Odyssey imaging system (Li-Cor Bioscience). RNA oligonucleotides used in RNA EMSA are listed in Supplementary Experimental Procedures.

**Fluorescence assays.** For Immunofluorescence (IF), HEK293T cells were fixed at 24 h post transfection, whereas A594 cells were fixed at 6 h post infection using fixation buffer (11% formalin in PBS) at room temperature for 10 min. Permeabilized slides were incubated with primary antibodies (SF2: (Santa Cruz, sc-33652, dilution 1:25; NS1 a kind gift provided by Dr Yee-Joo Tan of National University of Singapore, dilution 1:200) against proteins of interest diluted in blocking buffer at 37 °C for 1 h. After three washes in buffer (0.05% Tween 20 in PBS), Alexa Fluor-conjugated secondary antibodies (Abcam) diluted in wash buffer were added evenly onto slides. For sequential immunofluorescence (IF) plus fluorescence *in situ* hybridization (FISH) assay, after staining with secondary antibodies, slides were washed in PBS and incubated in FISH wash buffer (10% Formamide, 300 mM NaCl, 30 mM sodium citrate pH 7.0) for 5 min at room temperature. RNA probes (Biosearch Technologies) for NS1 mRNA detection dissolved in hybridization buffer (10% Dextran sulphate, 10% formamide, 300 mM NaCl, 30 mM sodium citrate pH 7.0) were distributed on slides and incubated at 37 °C for 5 h. Slides for IF or FISH were mounted using mounting buffer with DAPI (VECTASHIELD). IF images were captured by LSM 700 confocal microscope ( × 63 oil-immersion lens) and those of FISH were acquired by wide field microscope ( × 60 oil-immersion lens). NS1-NG/SF2-CG and NG/CG plasmids for the bimolecular fluorescence complementation (BiFC) imaging assay were constructed as previously reported[60]. Green fluorescent protein signals emitted by BiFC were observed using a LSM 780 confocal microscope ( × 40 oil-immersion lens).

**RNA interference knockdown.** Oligos (SiRNA-1, -2, -3: HSS109654, HSS109655, HSS109656 and scrambled negative control: 12935300) were purchased from ThermoFisher Scientific. HEK293T cells were reverse transfected with each SiRNA using RNAiMAX (ThermoFisher Scientific) in accordance with the user manual for 24 h before use in subsequent experiments.

**Statistical analysis.** Student's *t*-test and two-way analysis of variance with the Bonferroni post test were performed using Graphpad Prism 7 software.

**Biosafety measures.** Experiments using H7N9 viruses or derivatives were conducted in BSL-3 facilities at the University of Hong Kong, following the Standard Protocols of the BSL-3 core facility of the University of Hong Kong, and government and institutional guidelines. Personnel that conducted these experiments were trained and evaluated according to the requirements of University of Hong Kong BSL-3 core facility. The animal studies were carried out in the BSL-3 facility of the University of Hong Kong, with approval from the Committee on the Use of Live Animals in Training and Research of the University of Hong Kong (CULATR: 3653-15). The CULATR follows Hong Kong legislation

and Association for Assessment and Accreditation of Laboratory Animal Care International recommended standards/guidelines (http://www.aaalac.org/about/guidelines.cfm).

**Data availability.** All relevant data are available from the authors upon request.

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

## Acknowledgements

This study was supported in part by the Research Grants Council of the Hong Kong SAR (7629/13M,17103214 and 17154516), the Health and Medical Research Fund (14131032), the Areas of Excellence Scheme of the University Grants Committee (Grant AoE/M-12/06), Guangdong Innovative and Entrepreneurial Research Team Program (Number 2014ZT05S136), National Natural Science Foundation of China (31670934), Larry Yung, Richard Yu, Carol Yu and the Providence Foundation Limited in memory of the late Dr Lui Hac Minh. We are grateful to Dr Jane Rayner for editing the manuscript.

## Author contributions

X.H., K.-Y.Y. and H.C. conceived the study and designed the experiments. X.H., M.Z., P.W., B.W.-Y.M., S.L., S.-Y. L., P.C., Y.-C.L., H.L., Y.-C.C. and W.S. performed experiments. X.H. and H.C. analysed data. X.H. and H.C. wrote the paper.

## Additional information

**Competing financial interests:** The authors declare no competing financial interests.

