## [Peer Review File · Nature Communications]

Reviewers' comments:

Reviewer #1 (Remarks to the Author):

Huang et al have discovered a new way in which the splicing of influenza virus segment 8 mRNA is regulated. They discovered this by first looking the recently emerged H7N9 viruses, that show a replication advantage in mammalian cells that correlated with a sequence variation in the NS RNA. This mutation might either have acted via its effect on the coding of NS1 protein (it changed an amino acid), or by affecting the relative splicing of NS1vs NEP mRNAs since it was located within a potential ESE site. They demonstrate that this ESE site is used in a synthetic RNA, and also that the mutation G540A affects NEP levels and the ration of NS1 to NEP mRNAs during virus infection.

Next they use some really nice biochemistry to show in EMSA an interaction between SF2 and the newly described ESE in RNA targets. Weaker interaction correlates with weaker splicing of the H7N9 wt NS mRNA.

They also introduce the concept the NS1 itself regulates the interaction between SF2 and the ESE, by showing an interaction between NS1 and SF2 that is dependent on NS1 amino acids 38 ad 41.

Finally they perform some in vivo experiment in mice and show that H7N9 virus driven weight loss is attenuated if the ESE is altered so that splicing is restored.

A lot of this work is very novel, and will be of interest to the virology community. However as it stands there are a lot of unsubstantiated claims and unfinished or unsupported data.

The authors several times make the point that the effect of the ESE mutation on virus replication is only apparent in mammalian cells and not in DF-1 chicken cells. This is curious since presumable avian cells possess an SF2 protein and regulate splicing in similar ways as mammalian cells? If the authors want to press the point about host species-specific effects, they really should show that the ratios of spliced and unspliced mRNAs is not different between wt and A540G mutant virus in DF-1 cells- This indeed would be very powerful evidence to support their hypothesis.

The authors present a series of experiment to show that the coding change in NS1 does not explain the difference in replication between wt and A540G mutant virus. Much of this work is done with overexpressed NS1 and the authors only show a single concentration of transfected plasmid that is likely not at all representative of the NS1 expression level in infected cells. The authors should titrate in lower levels of NS1 plasmid in figure 2A B and C to see if there are differences in NS1 potency in these assays. In addition the effect of NS1 n polymerase activity in figure 2A is surprisingly large, this data is normalized for polII driven expression but are the authors sure that what they are measuring here is a true effect on polymerase rather than the effect of both NS1 proteins decreasing polII drive luciferase signal? Raw data might reveal the answer here. Finally on this point, one wonders if the authors could not have more neatly shown effects are due to splicing efficiency and not NS1 amino acid change if they had generated further mutant altered in ESE at the RNA level without changing coding capacity.

Along these lines the extent of use of reverse genetic to make mutant viruses is

disappointing and some cases inadequate to justify the claims made. For example, in figure 1B, the authors first show data about H7N9 altered in ESE and then about H9N2. However a close reading of the description of the H9N2 virus reveals that the authors have only changed the HA and NA genes and are using exactly the same set of 6 internal genes including the NS gene segment as in the first experiment. The authors should be performing this second RG experiment in a completely different gene set that is entirely derived from a real H9N2 virus. Otherwise their figure 1B does not really show anything novel at all and should be removed.

Figure 2H makes nice use of a model system to study splicing. The results presented as a gel of RT-PCR products could be quantified, since it is not possible to see a difference in splicing efficacy between the wt and the A540G mutants? If it is not possible to see a difference in this system the authors should simply acknowledge that.

Figure 4A is quite confusing with so many SF2s all at once! Do the authors want to imply that at the acceptor site two SF2 proteins are bound at the same time to both the acceptor site and the ESE?

Figure 5E, why is the level of FLAG SF2 so low in input in presence of RNase A? The effects of NS1 on splicing make the story very complicated and the authors have tried to control for this. In case of the mutant NS1 at 38 and 41, have they shown how these changes affect the Nuclear localization of NS1 since surely this could easily explain the lack of activity and interaction with SF2. This should be added to figure S6.

The mouse work is insufficient as presented. Only weight loss is shown. It is not clear whether as in vitro, virus replication has been affected by the mutation in the ESE. Since weight loss in mice can be driven either by excess cytokine or increased virus, it would be important to assess viral loads in the lungs of infected mice. In the same vein the claim made in discussion line 347 cannot be made unless virological data is presented, a virus ability to infect humans is unlikely to be related to its ability to cause weight loss in a mouse, but increased replication might explain it.

Lines 317-320 in discussion are unclear and should be reworded. Is polymerase involved in M splicing or not? Supplementary figure S2B and 2C are presented the wrong way round as talked about in the text.

It is not acceptable to write a manuscript on segment 8 splicing that implies a role for the host in regulating this process without acknowledging the work from Naffakh and coauthors that host factors RED and SMU1 control segment 8 splicing by interacting with viral polymerase. In this regard a weakness of the current paper is that figure 2 D to H are very artificial systems in which the splicing of a novel segment 8 mRNA is measured out of context of virus infection with no polymerase proteins present.

The working model present in figure S7 is a good addition. But it is still not made clear

whether the replication difference between wt and mutant A540G is due to wt making less NEP and/or more NS1 protein? Manz et al presented data that NEP can rescue defects in avian polymerase in mammalian cells, but the H7N9 virus used here already has a mammalian adapted polymerase with E627K mutation in PB2? Does that imply it needs to decrease the impact of NEP by selecting changes in the ESE efficiency? The authors should make their views clearer in the discussion about the relative importance of more NS1 vs less /slower accumulation of NEP.

Line 99 should T539C be U539C as it is in RNA?

Line 144 typo gens should be genes

Line 121 typo 9H9N2) should be (H9N2)

Line 134 there are better reference to use to support the effect of NS1 on polymerase activity, and 2 of the ones used here support a role for NEP rather than NS1 - rephrase or choose different citations.

Line 247 figure S5 not S6.

Reviewer #2 (Remarks to the Author):

In their manuscript 'Characterization of a novel exonic splicing enhancer in the NS segment and regulation of 2 NS mRNA splicing of influenza A (H7N9) virus' Chen and colleagues investigate a connection between alternative splicing of the influenza A NS RNA and the host splicing machinery. While I think that this topic is very interesting, there are many problems with the manuscript that, in my opinion, render it unsuitable for publication in Nature Communications.

The authors start by identifying a base substitution (G540A) in the H7N9 virus. They then analyze viral propagation in wt and G540A strains. The base substitution is then suggested to be located in an ESE sequence and to alter the NEP/NS1 mRNA ratio. Experiments are then performed to suggest binding of SF2 to the ESE and an interaction between SF2 and NS1 protein is suggested to control this NS alternative splicing. Finally, experiments are performed to show an in vivo relevance of the base substitution.

Here are some of the problems I see:

- Why are the error bars in Figure 1B so small? Are these biological replicates? Or technical replicates? And why is the error biased towards one direction? Are the differences in Figure 1B significant?
- The splicing differences shown in Figure 2E and G and also 3C and D are very small. The ratio changes by the factor 2, change of the individual isoforms is much smaller (3C, D). In all these experiments the error is so small, I believe within the technical error of a qPCR machine, that I am wondering whether this is based on biological or technical replicates. There is no n mentioned for the experiments in Figure 2. What does n mean in the other figures (technical or independent biological replicates)?
- What do the authors make of the altered fold change in 2G (compare NS and NS-501-

stop)?

- Such small changes in isoform expression as seen here can easily be explained by slightly altered mRNA stability. The authors have to test mRNA stability of wt and mutant mRNAs to be able to suggest a change in splicing. The qPCR primer would amplify genomic DNA and plasmid DNA. Was a DNase digestion done and a -RT control?
- What is the upper band in 2H SMN2? And shouldn't there be a difference in this assay of the wt and A540G version?
 - The splicing change and the change in protein shown in 3A are not exactly consistent. In 2E there is a difference 12 h.p.i at the RNA level, whereas this is not the case on protein level (3A).
 - There is no n mentioned in 3B and there are no error bars. I therefore assume that this is n=1. The same goes for Figure 4D and 4E. It is impossible to draw any conclusions here.
 - 4B is based on SF2-Flag overexpressing NE. There should be endogenous SF2 binding to the RNA visible as well. What is the expression level of endogenous vs SF2-Flag?
 - Use full length SF2 for experiments in 4D and 4E. Use different amounts of competitor in 4E. Repeat and quantify at least 3 times. There may be overall less loaded in the right lane in 4E as the double band slightly above the free probe is also reduced.
 - In 4F show another (non-binding) RNA as control. Show untransfected cells and Flag-IP as another control.
 - 4G: If endogenous SF2 binds to the RNA, where is it in 4B?
 - There should be a difference in the RIP assays with wt and A540G. That should be shown.
 - None of the experiments in Fig. 4 shows direct binding of SF2 to the RNA (as stated in the text). This could all be indirect. Show UV X-link to confirm direct binding.
 - In 5A a crucial control is missing: V5 IP in cells that only express SF2-Flag, same in 5C.
 - The Flag input in 5D is so different and the intensity of the IP (Flag blot) basically follows the input, with reduced signal. No conclusion possible.
 - Is SF2-Flag gone in the input of 5E upon RNase treatment? Why?
 - If recombinant proteins can be detected by Coomassie in 5F, please do a pulldown and show the interaction and the Gst control on one Coomassie gel.
 - The IF colocalization studies in the supplement are not convincing at all.
 - Figure 6 should also contain the A540G mutant. To show a connection to SF2, there should be coexpression of SF2 and NS1 analyzed or knock down of SF2 and expression of NS1.
 - If the effect of NS1 goes through SF2 and the ED part does not bind SF2 anymore (suggested in 5B), why has the ED domain still an effect on the NEP/NS1 ratio in 6C?
 - In my opinion, the number of mice in Figure 7 is too small to base conclusions on it.

Reviewer #3 (Remarks to the Author):

The manuscript by Huang et al reports on a novel exonic splicing enhancer present in the NS segment of the H7N9 influenza virus. This enhancer alters the splicing pattern of the NS segment, which generates the NS1 and NS2 (NEP) mRNAs. A G540A substitution in this enhancer sequence increases H7N9 replication in mammalian cells and maintains replication in avian cells. The authors then showed that the enhancer binds the cellular splicing regulator SF2, which interacts with the viral NS1 protein. NS1 has inhibitory function in the

splicing reaction. Thus, these interactions allow proper expression of NS1 and NS2 proteins at specific times during the infection cycle. These findings are important for understanding cross species transmission of influenza viruses as well as the role of alternative splicing mechanisms in this process.

There are a few additional points that need to be addressed:

1. In the experiments showing SF2 interaction with NS RNA (Fig. 4D and E), the authors refer to changes in affinity without showing quantitative experiments with several different concentrations. These should be performed in a more quantitative manner.

2. Since RNA-IP was performed without crosslinking, additional controls are necessary to determine specificity. For example, in addition to the NS RNA the authors should check whether other viral RNAs are also interacting, particularly the ones that are not spliced. Since viral RNAs are abundant, one concern here is non-specific binding.

3. Regarding the sentence on Line 180: "Interestingly, M1 protein expression was also affected by A540G mutation, presumably due to the altered level of NS1 as reported in another study 9." A new paper on this topic has been recently published by Mor et al in Nature Microbiology (Nat Microbiol. 2016;2016. pii: 16069. Epub 2016 May 27). This should be discussed and cited here.

4. Regarding the sentence starting on line 231: "NS1 inhibits host antiviral responses through suppression of the host mRNA splicing process....." This sentence needs to be re-written as NS1 is a multifunctional protein that inhibits host antiviral response by several mechanisms, including splicing.

5. Regarding the sentence starting on line 258: "The NS1 protein regulates the host splicing process by interacting with nuclear small RNA to prevent the formation of spliceosome complexes and blocks nucleocytoplasmic transport of host mRNAs to intervene with expression of genes required for host antiviral functions." References related to the effects of NS1 on nucleocytoplasmic transport should be added, including Qiu Y and Krug RM (J. Virol. 1994) and Satterly et al (PNAS 2007).

Point by point response to reviewers:

Reviewer #1 (Remarks to the Author):

Huang et al have discovered a new way in which the splicing of influenza virus segment 8 mRNA is regulated. They discovered this by first looking the recently emerged H7N9 viruses, that show a replication advantage in mammalian cells that correlated with a sequence variation in the NS RNA. This mutation might either have acted via its effect on the coding of NS1 protein (it changed an amino acid), or by affecting the relative splicing of NS1 vs NEP mRNAs since it was located within a potential ESE site. They demonstrate that this ESE site is used in a synthetic RNA, and also that the mutation G540A affects NEP levels and the ratio of NS1 to NEP mRNAs during virus infection. Next they use some really nice biochemistry to show in EMSA an interaction between SF2 and the newly described ESE in RNA targets. Weaker interaction correlates with weaker splicing of the H7N9 wt NS mRNA.

They also introduce the concept the NS1 itself regulates the interaction between SF2 and the ESE, by showing an interaction between NS1 and SF2 that is dependent on NS1 amino acids 38 ad 41. Finally they perform some in vivo experiment in mice and show that H7N9 virus driven weight loss is attenuated if the ESE is altered so that splicing is restored. A lot of this work is very novel, and will be of interest to the virology community. However as it stands there are a lot of unsubstantiated claims and unfinished or unsupported data.

The authors several times make the point that the effect of the ESE mutation on virus replication is only apparent in mammalian cells and not in DF-1 chicken cells. This is curious since presumable avian cells possess an SF2 protein and regulate splicing in similar ways as mammalian cells? If the authors want to press the point about host species-specific effects, they really should show that the ratios of spliced and unspliced mRNAs is not different between wt and A540G mutant virus in DF-1 cells- This indeed would be very powerful evidence to support their hypothesis.

Response: The authors agree with this reviewer's argument and have performed experiments to compare the levels of spliced and unspliced mRNAs using DF-1 cells infected with WT or A540G virus. As shown in Figure S3, while A540G substitution also slightly increased splicing efficiency of NS mRNA in DF-1 cells, the extent is much less than that observed in mammalian cells. This new evidence further supports our hypothesis that 540A is associated with host adaptation, enabling efficient virus replication in mammalian cells but not attenuating virus replication ability in avian hosts. These results have been included in the revised version (Fig. S3c, lines 154-156)

The authors present a series of experiment to show that the coding change in NS1 does not

explain the difference in replication between wt and A540G mutant virus. Much of this work is done with overexpressed NS1 and the authors only show a single concentration of transfected plasmid that is likely not at all representative of the NS1 expression level in infected cells. The authors should titrate in lower levels of NS1 plasmid in figure 2A B and C to see if there are differences in NS1 potency in these assays. In addition the effect of NS1 n polymerase activity in figure 2A is surprisingly large, this data is normalized for polII driven expression but are the authors sure that what they are measuring here is a true effect on polymerase rather than the effect of both NS1 proteins decreasing polII drive luciferase signal? Raw data might reveal the answer here. Finally on this point, one wonders if the authors could not have more neatly shown effects are due to splicing efficiency and not NS1 amino acid change if they had generated further mutant altered in ESE at the RNA level without changing coding capacity.

Response: To address this concern, the authors have performed experiments to test the effect of different doses of NS1 on reporter activity and confirmed that both WT and A540G increase polymerase activity while suppressing the interferon promoter. Raw data showed that the enhancement or suppression effect was not caused by any effect on the normalizer (Renilla reporter). The A540G mutation is a reverse substitution of a natural substitution; we have tried to test more mutations within the ESE site but have found it to be almost impossible to introduce mutations which may influence splicing efficiency, based on bioinformatics calculations, while not changing amino acids in either the NS1 or NEP proteins. We assume that 540A is naturally selected for fitness of H9N2 virus. (Figure 2a& b, lines 131-132, 134).

Along these lines the extent of use of reverse genetic to make mutant viruses is disappointing and some cases inadequate to justify the claims made. For example, in figure 1B, the authors first show data about H7N9 altered in ESE and then about H9N2. However a close reading of the description of the H9N2 virus reveals that the authors have only changes the HA and NA genes and are using exactly the same set of 6 internal genes including the NS gene segment as in the first experiment. The authors should be performing this second RG experiments in a completely different gene set that is entirely derived from a real H9N2 virus. Otherwise their figure 1B does not really show anything novel at all and should be removed.

Response: The reviewer's point is well taken and we have performed experiments using RG virus completely derived from an H9N2 strain (A/HongKong/308/2014). Similar to what we have observed in rH9N2 virus, A540G mutant H9N2 virus showed lower growth rates compared to the WT counterpart in mammalian cells (A549) but not in avian cells (DF-1). (Figure 1b-H9N2, lines 111-113, 115).

Figure 2H makes nice use of a model system to study splicing. The results presented as a gel of RT-PCR products could be quantified, since it is not possible to see a difference in splicing efficacy between the wt and the A540G mutants? If it is not possible to see a difference in this system the authors should simply acknowledge that.

Response: This experiment is to prove that the putative ESE identified in this study can be functional for splicing in another gene setting. The agarose gel analysis presented is not suitable for quantifying the difference between H7-WT and A540G mutant. We have added a statement to this effect in the revised version (Figure 2h, lines 169-170).

Figure 4A is quite confusing with so many SF2s all at once! Do the authors want to imply that at the acceptor site two SF2 proteins are bound at the same time to both the acceptor site and the ESE?

Response: Figure 4A is an illustration of the regions from which the probes used in the experiment were derived. To test the binding of SF2 to the identified ESE site, this study also included several probes derived from relevant regions which potentially can be bound by SF2. We have modified the illustration to allow easier understanding of ESE SF2 binding site (Figure 4a).

Figure 5E, why is the level of FLAG SF2 so low in input in presence of RNase A?

Response: We realized that some proteins became unstable and were precipitated in our previous experimental system. We have re-done the experiment using a new protocol to avoid protein precipitation. In the new experiment, an RNA-mediated protein-protein interaction is included as a positive control (revised Figure 5e, lines 254-257).

The effects of NS1 on splicing make the story very complicated and the authors have tried to control for this. In case of the mutant NS1 at 38 and 41, have they shown how these changes affect the Nuclear localization of NS1 since surely this could easily explain the lack of activity and interaction with SF2. This should be added to figure S6.

Response: We have included additional data showing that mutant R38A/K41A NS1 still localizes to the nucleus, like WT NS1. Therefore, the loss of interaction between mutant R38A/K41A and SF2 is not caused by a change of cellular localization (Figure S6a bottom panel, lines 287-292).

The mouse work is insufficient as presented. Only weight loss is shown. It is not clear whether as in vitro, virus replication has been affected by the mutation in the ESE. Since weight loss in mice can be driven either by excess cytokine or increased virus, it would be important to assess viral loads in the lungs of infected mice. In the same vein the claim made in discussion line 347 cannot be made unless virological data is presented, a virus ability to infect humans is unlikely to be related to its ability to cause weight loss in a mouse, but increase replication might explain it.

Response: We have further examined virus titers in lung tissues collected from infected animals 72 hours post infection. In the groups challenged with viral doses of 4.75×10^3 and 4.75×10^4 PFU, no significant difference in virus titers was observed between WT- and A540G-infected mice. In the highest dose (4.75×10^5 PFU) challenge group, A540 virus infected mice had higher

virus titers in the lung. These data seem to point to the possibility that more significant body weight loss and mortality rates may be caused by the different properties of the two viruses, such as lower levels of expression of NS1 which may attenuate the pathogenicity of the A540G mutant virus since NS1 is a key virulent element for influenza A virus. The primary purpose of this animal experiment is to test the phenotypic properties of WT and A540G viruses in an *in vivo* system. We have modified the description in the revised version (Figure S7, lines 306-309).

Lines 317-320 in discussion are unclear and should be reworded. Is polymerase involved in M splicing or not? Supplementary figure S2B and 2C are presented the wrong way round as talked about in the text.

Response: We have modified the text from lines 317-320, and the correction has been made for Figure S2b and 2c (Lines 331-335).

It is not acceptable to write a manuscript on segment 8 splicing that implies a role for the host in regulating this process without acknowledging the work from Naffakh and coauthors that host factors RED and SMU1 control segment 8 splicing by interacting with viral polymerase. In this regard a weakness of the current paper is that figure 2 D to H are very artificial systems in which the splicing of a novel segment 8 mRNA is measured out of context of virus infect with no polymerase proteins present.

Response: The reference in question was cited in the previous version, and we have discussed the work of Naffakh and coauthors in the revision. We agree with this reviewer that viral polymerase is also involved in the regulation of NS mRNA splicing, and the revised discussion now mentions the role of viral polymerases in regulation of NS splicing by RED and SMU1 (Lines 358-361) .

To confirm the effect of NS-A540G on NS mRNA splicing we have conducted both co-transfection of pHI PLASMID with viral RNP to transcribe NS mRNA (Figure 2g) and virus infections (Figure 2e); both of these systems have polymerase present.

The working model present in figure S7 is a good addition. But it is still not made clear whether the replication difference between wt and mutant A540G is due to wt making less NEP and/or more NS1 protein? Manz et al presented data that NEP can rescue defects in avian polymerase in mammalian cells, but the H7N9 virus used here already has a mammalian adapted polymerase with E627K mutation in PB2? Does that imply it needs to decrease the impact of NEP by selecting changes in the ESE efficiency? The authors should make their views clearer in the discussion about the relative importance of more NS1 vs less /slower accumulation of NEP.

Response: We have modified Figure S7 (previous version) to explain the regulation of NEP/NS1 expression by cis and trans mechanisms more clearly (Revised Figure S8 and lines 367-369).

Line 99 should T539C be U539C as it is in RNA?

Response: The correction has been made.

Line 144 typo gens should be genes

Response: The correction has been made in the revised version.

Line 121 typo 9H9N2) should be (H9N2)

Response: The correction is made in the revised version

Line 134 there are better reference to use to support the effect of NS1 on polymerase activity, and 2 of the ones used here support a role for NEP rather than NS1 - rephrase or choose different citations.

Response: The reviewer's points are well taken and corrections have been made accordingly in the revised version.

Line 247 figure S5 not S6.

Response: The mistake has been corrected.

Reviewer #2 (Remarks to the Author):

In their manuscript, 'Characterization of a novel exonic splicing enhancer in the NS segment and regulation of 2 NS mRNA splicing of influenza A (H7N9) virus' Chen and colleagues investigate a connection between alternative splicing of the influenza A NS RNA and the host splicing machinery. While I think that this topic is very interesting, there are many problems with the manuscript that, in my opinion, render it unsuitable for publication in Nature Communications.

The authors start by identifying a base substitution (G540A) in the H7N9 virus. They then analyze viral propagation in wt and G540A strains. The base substitution is then suggested to be located in an ESE sequence and to alter the NEP/NS1 mRNA ratio. Experiments are then performed to suggest binding of SF2 to the ESE and an interaction between SF2 and NS1 protein is suggested to control this NS alternative splicing. Finally, experiments are performed to show an in vivo relevance of the base substitution.

Here are some of the problems I see:

- Why are the error bars in Figure 1B so small? Are these biological replicates? Or technical replicates? And why is the error biased towards one direction? Are the differences in Figure 1B significant?

Response: These results are all from biological replicates of three or more. To assure that our results are reliable, we have re-done one of the rH9N2 virus infections of A549 cells and the result was consistent with the previous one. All errors were calculated from at least three experiments using the methods described in Methods and Materials. In the revision, we have included additional results with a pair of RG viruses containing all of the segments from an H9N2 virus (A/HongKong/308/2014). The results in Figure 1B have been statistically analyzed and a significant difference between WT and A540G mutant virus in terms of growth ability in A549 cells, but not in DF-1 cells, was observed (Figure 1b and lines 111-115).

- The splicing differences shown in Figure 2E and G and also 3C and D are very small. The ratio changes by the factor 2, change of the individual isoforms is much smaller (3C, D). In all these experiments the error is so small, I believe within the technical error of a qPCR machine, that I am wondering whether this is based on biological or technical replicates. There is no n mentioned for the experiments in Figure 2. What does n mean in the other figures (technical or independent biological replicates)?

Response: The data represent biological replicates. It is our standard practice to repeat experiments at least three times to calculate the standard deviation (SD). Results of Figure 2 were from three experiments. (Revised Figure 2 and legends).

- What do the authors make of the altered fold change in 2G (compare NS and NS-501-stop)?

Response: This experiment is to assess the effect of NS1 on suppression of NEP/NS1 mRNA splicing efficiency by expressing different truncated NS1 proteins from NS segments carrying either 540A (WT) or 540G (mutant). Because NS1 has a general inhibitory effect on splicing of NS mRNA, expression of various truncated forms of NS1 tended to increase NEP/NS1 ratios compared to expression of WT full length NS1. However, the A540G mutant versions of NS1 consistently resulted in higher NEP/NS1 mRNA ratios than WT versions, supporting the contention that the effect of A540G on splicing is independent of expression of NS with 172E or 172K.

- Such small changes in isoform expression as seen here can easily be explained by slightly altered mRNA stability. The authors have to test mRNA stability of wt and mutant mRNAs to be able to suggest a change in splicing. The qPCR primer would amplify genomic DNA and plasmid DNA. Was a DNase digestion done and a -RT control?

Response: We have tested the stability of WT and mutant mRNAs and found there to be no apparent difference in stability of NS1 and NEP mRNAs between the WT and A540G mutant. DNase digestion was included and the absence of residual DNA contamination verified for all RT-qPCR assays in transfection experiments throughout the study. (Figure S3a & S3b and lines 190-191,194)

- What is the upper band in 2H SMN2? And shouldn't there be a difference in this assay of the wt and A540G version?

Response: This band also appeared in the journal article which our experiment is based on. The authors have performed cloning and sequencing to characterize the identity of this DNA band and found that it is a hybrid of two isoforms which may form a loop and cause slower migration on the agarose gel. The DNA agarose gel analysis of PCR products presented here cannot be used for quantitative analysis.

- The splicing change and the change in protein shown in 3A are not exactly consistent. In 2E there is a difference 12 h.p.i at the RNA level, whereas this is not the case on protein level (3A).

Response: To reconfirm the result of this experiment, we have repeated the experiment during the revision process and the new result confirms the difference in NEP/NS1 protein levels between WT and A540G mutant in the infection setting. Our result shows that A540G increases NEP expression, particularly in the earlier hours of infection (Revised Figure 3a & b).

- There is no n mentioned in 3B and there are no error bars. I therefore assume that this is n=1. The same goes for Figure 4D and 4E. It is impossible to draw any conclusions here.

Response: Figure 3A is representative of the three biological replicates performed the protein bands from Western blots were individually quantified using Image J and the results displayed in

Figure 3B in the revised version. The results clearly show the effect of A540G substitution on NEP/NS1 mRNA ratio and protein levels in infection.

Both 4D and 4E were repeated three times and the results showed similar trends. Levels of band intensity were quantified from one of the three experiments (Revised Figure 4d & e).

- 4B is based on SF2-Flag overexpressing NE. There should be endogenous SF2 binding to the RNA visible as well. What is the expression level of endogenous vs SF2-Flag?

Response: The expression of endogenous SF2 is too low to be used in the gel shift assay. Based on our western blot analysis, the expression level of endogenous SF2 is more than 10-fold lower than that of SF2-Flag.

- Use full length SF2 for experiments in 4D and 4E. Use different amounts of competitor in 4E. Repeat and quantify at least 3 times. There may be overall less loaded in the right lane in 4E as the double band slightly above the free probe is also reduced.

Response: We have taken the suggestion from this reviewer and used full length SF2 and different amounts of SF2 in three repeated RNA EMSA experiments. Results are shown in the revised version (Revised Figure 4d & e).

- In 4F show another (non-binding) RNA as control. Show untransfected cells and Flag-IP as another control.

Response: Nuclear extracts from untransfected cells (NT) or Flag-IP were included in the RNA precipitation assay. Besides NS mRNA, non-binding U87 scaRNA and viral PB1 mRNA were used as controls in the experiments and the resulting data is presented in the revised figures (Revised Figure 4g & h and lines 225 & 228).

- 4G: If endogenous SF2 binds to the RNA, where is it in 4B?

Response: EMSA requires much more protein than an RNA ChIP assay. We can only demonstrate binding of endogenous SF2 using an RNA ChIP assay as shown in Figure 4G & H. Additional controls are included in the revision (Figure 4g & h).

- There should be a difference in the RIP assays with wt and A540G. That should be shown.

Response: As we have shown in the EMSA assay (Figure 4A and C), the splicing acceptor site also contains an SF2 binding site, close to the ESE site of NS mRNA. We consider that RIP will not be able to differentiate between WT and A540G mutant, because SF2 would bind to both the acceptor site and the NEP ESE site in the RNA ChIP assay. Therefore, we did not compare differential binding of WT and A540G in the RIP assay.

- None of the experiments in Fig. 4 shows direct binding of SF2 to the RNA (as stated in the text). This could all be indirect. Show UV X-link to confirm direct binding.

Response: To address this question, we have adopted another approach by purifying expressed SF2 protein through immunoprecipitation. Purified SF2 protein is shown on a protein gel to demonstrate purity (Figure 4F of revised version). Using this purified SF2, we repeated the EMSA and confirmed the binding of SF2 to the NEP ESE motif. This evidence indicates that SF2 directly binds the NEP ESE site in the NS mRNA from influenza virus, and not through interaction with other proteins (Revised Figure 4f and lines 217-220).

- In 5A a crucial control is missing: V5 IP in cells that only express SF2-Flag, same in 5C.

Response: We have repeated both experiments and included vector transfected controls as suggested by the reviewer (Revised Figure 5a & c).

- The Flag input in 5D is so different and the intensity of the IP (Flag blot) basically follows the input, with reduced signal. No conclusion possible.

Response: We have repeated this result with similar levels of input NS1 WT and mutant proteins, confirming that RBD is required for interaction with SF2 and that the 38/41A dual mutant abolishes the interaction, even though it is localized in the nucleus (Revised Figure 5d, Figure S6a).

- Is SF2-Flag gone in the input of 5E upon RNase treatment? Why?

Response: We realized that it was caused by the precipitation of protein after RNase treatment. To avoid such problems, we have repeated the experiment with a new experimental protocol and did not encounter the same problem. To further demonstrate the effect of RNase treatment, we have included a control showing that RNase A treatment abolishes the interaction between NS1 and RNA Helicase (Revised Figure 5e).

- If recombinant proteins can be detected by Coomassie in 5F, please do a pulldown and show the interaction and the Gst control on one Coomassie gel.

Response: The amount of GST pull down protein (NS1) is low and could not be easily detected by Coomassie blue staining. We have used western blotting to verify the pull down protein. This experiment confirms a direct interaction between the NS1 and SF2 proteins.

- The IF colocalization studies in the supplement are not convincing at all.

Response: The NS1 protein (green) is diffused in the nucleus and it is a little difficult to see spots of co-localization clearly. To support the contention of an interaction between NS1 and SF2 in

cells, we have used another imaging assay, BiFC, to show the association of NS1 and SF2 in transfected cells, in which green fluorescence is only emitted when NS1-NG and SF2-CG are associated, as shown in other studies (Joseph N. Hemerka et al., 2009 J Virol. and Satoshi Kakugawa et al., 2009 J Virol.) (Revised Figure S5c and lines 2591-260).

- Figure 6 should also contain the A540G mutant. To show a connection to SF2, there should be coexpression of SF2 and NS1 analyzed or knock down of SF2 and expression of NS1.

Response: The purpose of this experiment is to further verify an interaction between NS1 and SF2. We consider that substitution or variation at position 540 is not a concern here and that only a WT NS segment should be used in the experiment. We have performed SF2 knockdown in HEK293T cells and tested the effect of NS1 on the NEP/NS1 mRNA ratio. Using a NS-null replicon system, we found that restoring NS1 expression has a lesser inhibitory effect on splicing of NEP mRNA in cells where SF2 is efficiently knocked down, as demonstrated by a significantly increased ratio of NEP/NS1 mRNA (Revised Figure S6b and lines 285-292).

- If the effect of NS1 goes through SF2 and the ED part does not bind SF2 anymore (suggested in 5B), why has the ED domain still an effect on the NEP/NS1 ratio in 6C?

Response: While NS1 can regulate NEP/NS1 mRNA splicing through interaction with SF2, we cannot exclude other mechanisms which may also be involved in regulation of NEP/NS1 splicing, such as those suggested by other studies, for instance, RED-SMU1, which also interacts with NS1. It is possible that other host factors may interact with the ED domain to regulate NEP/NS1 mRNA splicing. Further studies are required to reveal SF2 independent regulation mechanisms.

- In my opinion, the number of mice in Figure 7 is too small to base conclusions on it.

Response: The experiment started with 7 mice in each group. Three mice were sacrificed on day three post infection for titrating virus titers in lung tissues. While it would be desirable to have more mice in each group to calculate the body weight and survival, we found that the changes in body weights for mice in each group were relatively consistent, as can be seen from the small error bars. Four mice in each group is sufficient to calculate MLD₅₀ (lines 306-309)

Reviewer #3 (Remarks to the Author):

The manuscript by Huang et al reports on a novel exonic splicing enhancer present in the NS segment of the H7N9 influenza virus. This enhancer alters the splicing pattern of the NS segment, which generates the NS1 and NS2 (NEP) mRNAs. A G540A substitution in this enhancer sequence increases H7N9 replication in mammalian cells and maintains replication in avian cells. The authors then showed that the enhancer binds the cellular splicing regulator SF2, which interacts with the viral NS1 protein. NS1 has inhibitory function in the splicing reaction. Thus, these interactions allow proper expression of NS1 and NS2 proteins at specific times during the infection cycle. These findings are important for understanding cross species transmission of influenza viruses as well as the role of alternative splicing mechanisms in this process.

There are a few additional points that need to be addressed:

1. In the experiments showing SF2 interaction with NS RNA (Fig. 4D and E), the authors refer to changes in affinity without showing quantitative experiments with several different concentrations. These should be performed in a more quantitative manner.

Response: We have re-done the experiment with different amounts of SF2. The results showed that A540G has higher affinity, forming RNA-protein complexes at lower concentrations of SF2 in the gel-shift assay. Similarly, different amounts of competitors were used in the shift assay and A540G again showed higher binding efficiency than the WT probe (Revised Figure 4d & e).

2. Since RNA-IP was performed without crosslinking, additional controls are necessary to determine specificity. For example, in addition to the NS RNA the authors should check whether other viral RNAs are also interacting, particularly the ones that are not spliced. Since viral RNAs are abundant, one concern here is non-specific binding.

Response: The reviewer's point is well taken and we have included additional cellular and viral RNA controls in the assay. The results are included in the revised version (Figure 4g & h and line 228).

3. Regarding the sentence on Line 180: "Interestingly, M1 protein expression was also affected by A540G mutation, presumably due to the altered level of NS1 as reported in another study 9." A new paper on this topic has been recently published by Mor et al in Nature Microbiology (Nat Microbiol. 2016;2016. pii: 16069. Epub 2016 May 27). This should be discussed and cited here.

Response: We have included the reference and discussed it in the revised version (Lines 182-183).

4. Regarding the sentence starting on line 231: "NS1 inhibits host antiviral responses through suppression of the host mRNA splicing process....." This sentence needs to be re-written as NS1 is a multifunctional protein that inhibits host antiviral response by several mechanisms, including splicing.

Response: A revision has been made to address this point (Line 237).

5. Regarding the sentence starting on line 258: "The NS1 protein regulates the host splicing process by interacting with nuclear small RNA to prevent the formation of spliceosome complexes and blocks nucleocytoplasmic transport of host mRNAs to intervene with expression of genes required for host antiviral functions." References related to the effects of NS1 on nucleocytoplasmic transport should be added, including Qiu Y and Krug RM (J. Virol. 1994) and Satterly et al (PNAS 2007).

Response: These references are included in the revised version (revised references 17, 42 and 43).

Reviewers' comments:

Reviewer #1 (Remarks to the Author):

The authors have addressed adequately many of the concerns raised however the following points still need attention:

1. In figure 5E a new protocol has been used for the precipitation assay but this is not described in the revised paper methods section. This is very important since the results are now quite different than were achieved with the old method.
2. The main issue now is the discrepancy between the effect of the G and A nucleotide in vitro and in vivo. In mice the virus with A540 replicates to lower titre, but induces more pathology. It would be very important to a) show cytokine response in the lungs of these mice to explain this and b) revise the title of the figure because the G540A clearly does NOT enhance virus fitness in vivo but rather enhances virulence.

Reviewer #2 (Remarks to the Author):

In the revised version of the manuscript 'Characterization of a novel exonic splicing enhancer in the NS segment and regulation of 2 NS mRNA splicing of influenza A (H7N9) virus' Chen and colleagues have addressed many of my initial concerns. This manuscript has improved substantially. I still have a few points that should be addressed to make this manuscript stronger:

Figure 1: Could you add an explanation why the A540G substitution does not have a coding effect on the NEP protein?

Figure 2: Panel H is the only experiment in the paper that shows the ESE to be sufficient for splicing regulation in a heterologous context. It would be important and make the manuscript much stronger, if the authors could use a quantitative assay and see a difference between the 540A/G versions. This experiment is crucial to support the model. A radioactive, low-cycle RT-PCR would be best, an isoform specific qPCR could work as well.

In addition, the higher band in the SMN2 background should be explained with the respective citation in the figure legend.

Figure 3: It is somewhat difficult to explain different effects of the viruses in vivo after several days of infection (figure 7), if the effect on protein expression is already gone after 12h. However, this result is consistent with no difference in viral titers after 3 days of infection. But this somehow questions the validity of the cell culture and in vitro data. This should at least be discussed.

Figure 4: Quantification of EMSAs have to contain errorbars. Alternatively, gels of the independent experiments can be shown in the supplement.

The labeling in panel g is not clear. I would replace 'IP' with 'transfected construct' or something like that.

The additional data in S3 is very interesting, but difficult to understand. If the authors want to make the point of adaptation, they have to explain why the 540A/G does not effect

splicing in the avian system. Could you show a WB of SF1 protein in the different cell lines?
On page 7, new part, line 5: Does that refer to Figure S3A? Please indicate.

Reviewer #3 (Remarks to the Author):

The authors have properly addressed my points. Therefore, I support publication in Nature Communications as the topic is novel and important for understanding cross species transmission and the role of splicing in this process.

Point by point response to reviewers:

Reviewer #1 (Remarks to the Author):

The authors have addressed adequately many of the concerns raised however the following points still need attention:

1. In figure 5E a new protocol has been used for the precipitation assay but this is not described in the revised paper methods section. This is very important since the results are now quite different than were achieved with the old method.

Response: We have included the protocol for the precipitation assay in the revised version.

2. The main issue now is the discrepancy between the effect of the G and A nucleotide in vitro and in vivo. In mice the virus with A540 replicates to lower titre, but induces more pathology. It would be very important to a) show cytokine response in the lungs of these mice to explain this and b) revise the title of the figure because the G540A clearly does NOT enhance virus fitness in vivo but rather enhances virulence.

Response: Our data showed no significant difference in virus titers in lung tissues from mice infected with 4.75×10^3 or 4.75×10^4 PFU of either WT or A540G virus. Mice infected with the highest virus dose (4.75×10^5 PFU) appeared very ill from day 2 post infection onwards, and we suspect that severe damage to the lung tissues was caused by the infection by day 3, with lung cells subsequently being unable to support further virus replication. Therefore, estimation of virus titers in the 4.75×10^5 PFU dose group cannot exactly reflect virus replication ability. We followed the reviewer's suggestion and analyzed IFN- β , TNF- α , IL-1 β and IL-6 mRNA levels in lung tissue homogenates, but found no significant difference between WT and A540G virus infected mice. However, we were not able to examine cytokine levels in lung fluids, a technique normally performed to detect aberrant cytokine expression in virus infected animals (such as ferrets), because mice are too small to conduct such experiment efficiently. The mechanism underlying the enhanced virulence associated with the substitution in the A540-NS genotype H7N9 virus needs to be further investigated and we have followed the reviewer's suggestion and revised the title of the figure and text accordingly (Revised Figure S7b and lines 312-319).

Reviewer #2 (Remarks to the Author):

In the revised version of the manuscript 'Characterization of a novel exonic splicing enhancer in the NS segment and regulation of 2 NS mRNA splicing of influenza A (H7N9) virus' Chen and colleagues have addressed many of my initial concerns. This manuscript has improved substantially. I still have a few points that should be addressed to make this manuscript stronger:

Figure 1: Could you add an explanation why the A540G substitution does not have a coding effect on the NEP protein?

Response: As illustrated in the revised version of Figure 1, A540G is a silent mutation in the NEP protein but causes K172E in the NS1 protein (Revised Figure 1a).

Figure 2: Panel H is the only experiment in the paper that shows the ESE to be sufficient for splicing regulation in a heterologous context. It would be important and make the manuscript much stronger, if the authors could use a quantitative assay and see a difference between the 540A/G versions. This experiment is crucial to support the model. A radioactive, low-cycle RT-PCR would be best, an isoform specific qPCR could work as well.

Response: We conducted quantitative RT-PCR to estimate the effect of A540G on splicing of the SMN gene and confirmed that A540G significantly enhances the efficiency of exon-7 inclusion (Revised Figure 2h and line 173).

In addition, the higher band in the SMN2 background should be explained with the respective citation in the figure legend.

Response: We have included an explanation for the higher band in the legend of the revised version of the figure (Figure 2h legend).

Figure 3: It is somewhat difficult to explain different effects of the viruses in vivo after several days of infection (figure 7), if the effect on protein expression is already gone after 12h. However, this result is consistent with no difference in viral titers after 3 days of infection. But this somehow questions the validity of the cell culture and in vitro data. This should at least be discussed.

Response: Changes in body weights and estimation of MLD_{50} of mice infected with WT and A540G mutant viruses clearly showed that WT virus is more virulent than the A540G mutant. However, virus titers in lung tissues collected on day 3 are not significantly different between WT and A540G mutant virus infected mice. We have further analyzed mRNA levels of cytokines (IFN- β , TNF- α , IL-1 β and IL-6) in lung homogenates and found there is no significant difference between WT and A540G virus infected mice in this regard. Since we only examined virus titers on day 3, we do not know if a difference may be observable at earlier time points. Further study is necessary to explore the mechanism underlying the enhanced virulence associated with altered splicing efficiency of NS (Revised Figure S7b and lines 312-319).

Figure 4: Quantification of EMSAs have to contain errorbars. Alternatively, gels of the independent experiments can be shown in the supplement.

Response: We have analyzed results from three separate experiments and conducted statistical analysis to include error bars to represent standard deviation in Figure 4 (Revised Figure 4).

The labeling in panel g is not clear. I would replace 'IP' with 'transfected construct' or something like that.

Response: The labels have been modified in the revised version (Revised Figure 4g).

The additional data in S3 is very interesting, but difficult to understand. If the authors want to make the point of adaptation, they have to explain why the 540A/G does not effect splicing in the avian system. Could you show a WB of SF1 protein in the different cell lines?

Response: We have included a Western blot showing SF2 expression in A549 and DF-1 cells in the revised version. As shown in the Western blot, avian cells (DF1) express very low levels of SF2 compared to mammalian cells, and it is therefore possible that avian cells are less sensitive to splicing changes caused by A540G substitution in the NS RNA because of this. However, further study is necessary to understand if splicing regulation by SF2 is more effective in mammalian than avian cells. (Revised Figure S3).

On page 7, new part, line 5: Does that refer to Figure S3A? Please indicate.

Response: The correction has been made.

REVIEWERS' COMMENTS:

Reviewer #1 (Remarks to the Author):

The authors have addressed the comments of the reviewers.
I appreciate the rewording of the title for the in vivo mouse figure but stress (for future) that mice are not too small to measure cytokines in lung fluids, many authors publish such data and in fact such work is not possible in ferrets due to lack of ferret specific reagents!

Reviewer #2 (Remarks to the Author):

The authors have dealt with my remaining concerns and I now support publication of their manuscript.